# FiDeLiS: Faithful Reasoning in Large Language Models for Knowledge Graph Question Answering

## Abstract

Large language models are often challenged by generating erroneous or 'hallucinated' responses, especially in complex reasoning tasks. To mitigate this, we propose a retrieval augmented reasoning method, FiDeLiS, which enhances knowledge graph question answering by anchoring responses to structured, verifiable reasoning paths. FiDeLiS uses a keyword-enhanced retrieval mechanism that fetches relevant entities and relations from a vector-based index of KGs to ensure high-recall retrieval. Once these entities and relations are retrieved, our method constructs candidate reasoning paths which are then refined using a stepwise beam search. This ensures that all the paths we create can be confidently linked back to KGs, ensuring they are accurate and reliable. A distinctive feature of our approach is its blend of natural language planning with beam search to optimize the selection of reasoning paths. Moreover, we redesign the way reasoning paths are scored by transforming this process into a deductive reasoning task, allowing the LLM to assess the validity of the paths through deductive reasoning rather than traditional logit-based scoring. This helps avoid misleading reasoning chains and reduces unnecessary computational demand. Extensive experiments demonstrate that our method, even as a training-free method which has lower computational costs and superior generality, outperforms established strong baselines across three datasets. The code of this paper will be released at https://anonymous.4open.science/r/FiDELIS-E7FC.

## 1 Introduction

The emergence and application of large language models (LLMs) (Brown et al., 2020; OpenAI, 2023; LLaMa-v3, 2024) have attracted widespread attention from researchers and the general public. They demonstrate remarkable reasoning capabilities, managing to solve complex reasoning problems through step-by-step thinking and planning (Wang et al., 2022b; Wei et al., 2022b). However, the reasoning of LLMs is not always reliable and may conflict with factual reality (Pan et al., 2024), limiting their application in areas requiring high reliability, such as healthcare (He et al., 2023) and science (Taylor et al., 2022).

Knowledge graphs (KGs) store high-quality knowledge in structured triplets, such as Wikidata (Vrandečić & Krötzsch, 2014), YAGO (Suchanek et al., 2007), and NELL (Carlson et al., 2010). KGs offer structured, explicit, and up-to-date factual knowledge, including domain-specific knowledge, providing a faithful knowledge source for reasoning. Moreover, each piece of information in KGs can be traced back to its source, providing context and provenance. This traceability not only aids in verifying the reliability of the information but also provides clear pathways of reasoning, making the interpretation process transparent. Due to its reliability and interpretability, it is considered as a promising method to improve the reliability of LLM reasoning. Therefore, several attempts have been conducted to integrate KGs with language models (Pan et al., 2024; Luo et al., 2024; Hu et al., 2023; Sun et al., 2023). Among them, knowledge graph question answering (KGQA) is a critical task to verify the effectiveness of incorporating the knowledge from KGs into reasoning models (He et al., 2021; Wang et al., 2023a; Yu et al., 2022). However, effectively integrating KGs into reasoning models for KGQA presents several challenges, including data sparsity, the complexity of query

interpretation, and the need for advanced inference capabilities. More specifically, we focus on two main questions concerning the integration of KGs with reasoning models as follows:

**(I) How to retrieve specific knowledge from KGs to allow more precise reasoning?** Existing solutions include direct retrieval (Sun et al., 2019; Jiang et al., 2022) and semantic parsing (Sun et al., 2020; Gu & Su, 2022). Direct retrieval utilize the query to find relevant knowledge triplets within the KGs. However, these triplets sometimes lack comprehensive semantic information due to variations in KG schema, such as entities being represented as machine identifiers (MIDs), rather than descriptive labels. This lack of semantic richness can obscure the relevance of the retrieved information with the user query, and becomes especially tricky in multi-hop question answering, where seemingly unrelated intermediate triplets may be pivotal in deriving the correct answer. On the other hand, semantic parsing aim to transform user queries into executable structured queries (e.g., SPARQL), which are then run on KGs. However, these methods often grapple with issues related to the non-executability or incorrect execution of the generated queries (Yu et al., 2022), which undermines the relaibility of these methods.

**(II) How to make the reasoning model understand and utilize the retrieved structured knowledge in KGs?** Existing solutions include (1) finetuning LMs to generate relation paths grounded by KGs. These reasoning paths are then used for LLMs to conduct faithful reasoning and generate interpretable results (Luo et al., 2024; Yu et al., 2022); (2) leverage the reasoning model to iteratively retrieve and reason over subgraphs from the KG, deciding at each step to either answer the question, or to continue the searching step (Sun et al., 2023; Gu et al., 2023; Jiang et al., 2022). However, for the first approach, the reasoning steps generated by language models are not guaranteed to exist in the KG, especially when multiple consecutive steps are combined into a reasoning path. We conduct error analysis in Section 3.3 and show that only 67% of generated reasoning steps are valid, while the remaining 33% of reasoning steps either have a format error or do not exist in the KG. The latter solution faces the challenge of determining the optimal stopping point for the exploration process. For example, ToG (Sun et al., 2023) prompts LLMs to assess the adequacy of current reasoning paths for answer generation. However, assessing the adequacy of reasoning paths is itself a complex task that demands a deep understanding of the domain. This lack of clarity can easily result in premature stopping or excessive continuation, further complicating the decision-making process in LLMs.

To address these two challenges, we propose a retrieval augmented reasoning method, FiDeLiS. It is designed to enhance KGQA by anchoring responses to structured, verifiable reasoning paths. FiDeLiS is composed of two major components, Path-RAG, which retrieves chain of entities and relations from KGs in a effective manner (discussed in Section 2.1), and Deductive-verification Beam Search (DVBS), which conducts deductive-reasoning-based beam search to generate multiple reasoning paths leading to final answers (discussed in Section 2.2).

The Path-RAG module uses a keyword-enhanced retrieval mechanism that fetches relevant entities and relations from a vector-based index of KGs to ensure high-recall retrieval. Compared to a vanilla retriever which easily returns outputs that are not useful for finding the correct answer based on relevance to the query, we incorporate LLMs to generate an exhaustive list of keywords from the query to maximize coverage and ensure that no potential reasoning paths will be overlooked. Once these entities and relations are retrieved, the module constructs candidate reasoning steps which are then refined using a stepwise beam search discussed below.

Next, DVBS leverages the *deductive reasoning* capabilities of LLMs (Ling et al., 2024; Huang & Chang, 2023) as a clear-defined criterion for automatically directing the beam search process step by step to create the complete reasoning paths. A key characteristic of DVBS is its integration of natural language planning (Song et al., 2023; Zhou et al., 2022) and beam search. This combination enhances the process of choosing the best reasoning paths by providing additional hints for decision making in LLMs. In addition, we redesign how reasoning paths are evaluated by converting the scoring process into a deductive reasoning task. This allows the LLM to validate the reasoning paths through deductive verification rather than traditional logit-based scoring. This proposed deductive verification serves as presie indicators for when to cease further reasoning, thus avoiding misleading reasoning chains and reducing the computational resources required. **Overall, our main contributions include:**

- We propose a retrieval augmented reasoning method, which enhances knowledge graph question answering by anchoring responses to structured, verifiable reasoning paths grounded by KGs.

- We propose a step-wise keyword retrieval method that enhances the recall of relevant intermediate knowledge from KGs. This ensures that all the paths we create can be confidently linked back to KGs, ensuring that they are faithful and reliable.

- We propose deductive verification as precise indicators for when to cease further reasoning, thus avoiding misleading the chains of reasoning and reducing unnecessary computation required.

- Extensive experiments show that our method, as a training-free method with lower computational cost and better generality, outperforms existing strong baselines in three datasets.

## 2 METHOD

**Notation.** *Definition 1.* A **reasoning step** is a pair $(r, e)$, where $r$ is the relation and $e$ is the corresponding entity. *Definition 2.* A **reasoning path** $\mathcal{P}$ is a pair $(s, \mathcal{T})$, where $s$ is the starting entity for the reasoning path, and $\mathcal{T}$ is a sequence of reasoning steps $\mathcal{T} = \{t_1, \ldots, t_n\}$ and $t_k = (r_k, e_k)$ denotes the $k$-th reasoning step in the path and $n$ denotes the length of the path. *Definition 3.* The **next-hop candidates** given path $\mathcal{P}$, denoted $\mathcal{N}_1(e_n)$, are defined as the 1-hop neighborhood of $e_n$, the last node in the reasoning path $\mathcal{P}$. *Definition 4.* A reasoning path $\mathcal{P} = (s, \mathcal{T})$ is **valid** if it is connected and each reasoning step $t_k = (r_k, e_k)$ is correct (in the sense that $(e_{k-1}, r_k, e_k)$ is a triplet in the KG, where $e_0$ is defined as $s$). For example, a valid reasoning path could be: $\mathcal{P} =$ Justin_Bieber $\xrightarrow{\text{people.person.son}}$ Jeremy_Bieber $\xrightarrow{\text{people.person.ex\_wife}}$ Erin_Wagner, which denotes that "Jeremy Bieber" is the father of "Justin Bieber" and "Erin Wagner" is the ex-wife of "Jeremy Bieber". Each entity and relation are linked in the corresponding KG.

**Overview.** The framework of FiDeLiS is as shown in Figure 1. Given a query, it first retrieves the reasoning step candidates from KGs (as discussed in Section 2.1). To ensure high recall, our method first generates a list of keywords from the query. And then based on the generated keywords, it retrieves entities and relations from an index (where the entities and relations are also embedded by vector representation obtained from an LM). Next, following the retrieved entities, $n$-hop reasoning step candidates are created and scored based on semantics similarity and neighborhoods aggregation. Then, a deductive verification-based beam search (discussed in Section 2.2) is employed to generate the top-$k$ reasoning paths leading to the answer. First, a natural language plan is generated from the query to provide more hints for LLMs decision making and then a beam search procedure is conducted to score each reasoning step. We redesign the scoring function by converting the process into a deductive reasoning task. This allows the LLM to validate connections between paths through simple verification rather than traditional logit-based scoring. This change helps prevent easily misleading reasoning chains and reduces the computational resources required.

### 2.1 PATH-RAG: REASONING PATH RETRIEVAL-AUGMENTED GENERATION

Path-RAG aims to retrieve reasoning path candidates from KGs and comprises three main steps: *initialization*, *retrieval*, *reasoning step candidates construction*, as depicted in Figure 1. The implementation details of each step are elaborated in the following paragraphs.

**Initialization.** We initiate the Path-RAG by generating embeddings for entities (nodes) and relations (edges) using a pre-trained language model (LM). We begin by extracting entities ($e_i$) and relations ($r_i$) from the KG, denoted as $\mathcal{E}$ and $\mathcal{R}$, respectively. Each entity and relation is then encoded using a pre-trained language model such as SentenceBert (Reimers & Gurevych, 2019) or E5 (Wang et al., 2022a), yielding dense vector representations:

$$z(e^i) = \text{LM}(e_i) \in \mathbb{R}^d, \ z(r^i) = \text{LM}(r_i) \in \mathbb{R}^d \tag{1}$$

where $d$ denotes the dimension of the output vector. These embeddings are stored in a nearest neighbor data structure, which facilitates quick retrieval of similar entities or relations based on their vector representations.

**Retrieval.** To retrieve the entities and relations that are relevant to the user query, the embeddings of them are utilized to populate a nearest neighbor index. Specifically, for a given query $q$, we use LLM to generate an exhaustive list of keywords or relation names referred to as $\mathcal{K}$ (the prompt is given in Section F). These keywords are generated to maximize the coverage of potential reasoning steps essential for answering the query. The list is derived as $\mathcal{K} = \text{LM}(\text{prompt}_p, q)$. Then each keyword

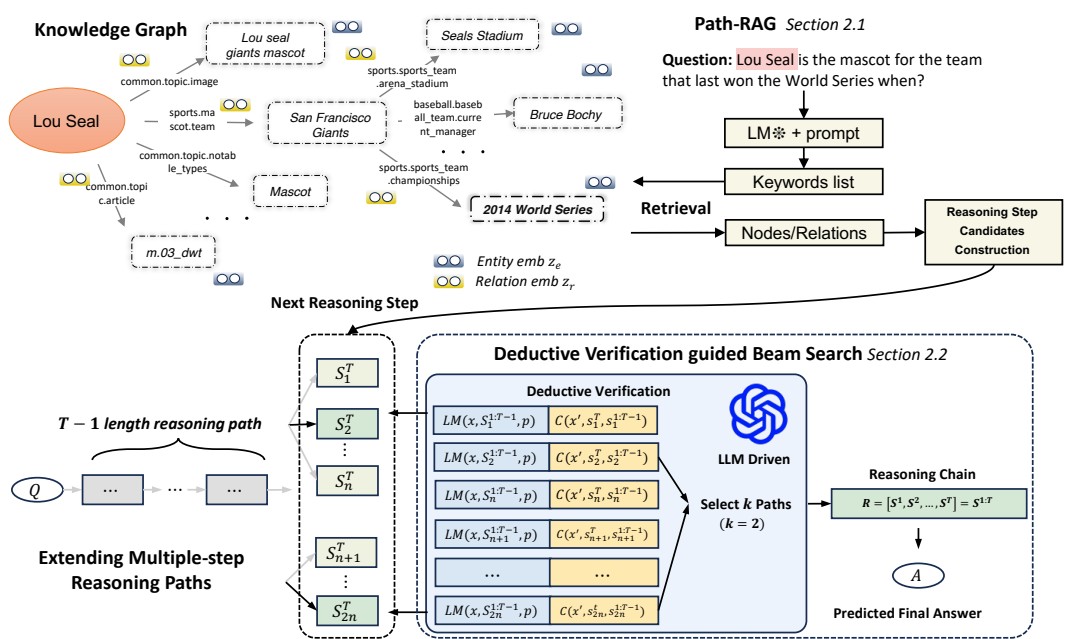

Figure 1: Overview of FiDeLiS. The goal of FiDeLiS is to extend the multiple-step reasoning paths $T$ grounded by KG. The figure shows given a query, Path-RAG first retrieve entities and relations that are relevant to the query, and construct the reasoning path candidates. Then DVBS constructs reasoning paths by iterative extending these path candidates using beam search. Specifically, at timestep $t$, DVBS leverage LLMs to choose the top-$k$ (here we set $k = 2$ as an example) reasoning steps and decide whether to continue the next search step or cease the reasoning path extension based on deductive verification. At last, FiDeLiS return the top-$k$ reasoning paths as the final output leading to the answers of the query.

$\in \mathcal{K}$ is concatenated into a string denoted as $K$ and then encoded into the same latent space by the same LM used in initialization step to obtain $z(K) = \text{LM}(K) \in \mathbb{R}^d$.

To find the entities and relations that best match the keyword embeddings, we compute the cosine similarity $\cos(\cdot, \cdot)$ between $z(K)$ and the embeddings of entities ($z(e)$) and relations ($z(r)$) in the KG. We then select the top-$m$ elements that exhibit the highest similarity:

$$\mathcal{E}_m = \text{argtopm}_{i \in \mathcal{E}} \cos\left(z(K), z(e)\right), \mathcal{R}_m = \text{argtopm}_{i \in \mathcal{R}} \cos\left(z(K), z(r)\right) \tag{2}$$

Each entity and relation in the retrieved set is then assigned a score, reflective of its similarity to the keyword embedding as $S_{\text{ent}}(e) = \cos\left(z(K), z(e)\right)$ and $S_{\text{rel}}(r) = \cos\left(z(k), z(r)\right)$.

**Reasoning Step Candidates Construction.** After the initial retrieval process, which runs once at the start of the algorithm, we iteratively construct reasoning step candidates to extend the reasoning paths based on entities $\mathcal{E}_m$ and relations $\mathcal{R}_m$ that are relevant to the given query. To guide the selection of these candidates, we propose another scoring function based on the derived relevance score $S_{\text{ent}}(e)$ and $S_{\text{rel}}(r)$. Our scoring function leverages triplets to represent the connectivity of KGs and assess whether extending a path in certain directions might lead to negative outcomes over multiple steps by incorporating next-hop neighbor's information.

$$S((r, e)) = S_{\text{rel}}(r) + S_{\text{ent}}(e) + \alpha \max_{\forall (r_j, e_j) \in N(e)} \left(S_{\text{rel}}(r_j) + S_{\text{ent}}(e_j)\right) \tag{3}$$

Here, $(r, e)$ refers to the reasoning step defined as a relation-entity pair. $S_{\text{rel}}(r_j)$ and $S_{\text{ent}}(e_j)$ are the similarity scores for relations and entities at the next step. The component $N(e)$ corresponds to entities reachable from $e$ within one hop. The factor $\alpha$ is used to balance short-term outcomes and long-term potential in reasoning paths: a higher $\alpha$ prioritizes paths with long-term benefits, even if they seem sub-optimal initially, while a lower $\alpha$ emphasizes immediate steps, potentially overlooking future impacts. Details of Path-RAG are provided in Algorithm 1.

## 2.2 DVBS: Deductive-Verification Guided Beam Search

DVBS is designed to prompt LLMs to iteratively execute beam search on the reasoning step candidates (provided by Path-RAG) to find the top-$k$ most promising reasoning paths that lead to the answer to the user query. At each timestep, we leverage the LLMs to choose the top-$k$ reasoning steps and decide whether to continue extending the reasoning paths or cease the extension process. It comprises three main steps: *planning*, *beam-search*, *deductive-verification*, as depicted in Figure 1. The implementation details of each step are elaborated in the following paragraphs.

**Plan-and-Solve.** Inspired by the recent works regarding planning capabilities of LLMs (Song et al., 2023; Zhou et al., 2022), we prompt LLM to generate the planning steps for answering the user query, denoted as $w$. This step aims to provide more hints for subsequent LLM decision making process. The detailed prompt can be found in Section F.

**Beam Search.** To enable multi-step reasoning, a reasoning path of $T$ steps is sequentially generated through several time steps as $\left[s^1, s^2, \ldots, s^t, s^T\right] = s^{1:T}$, where $s^t$ represents the reasoning step at timestamp $t$. We constrain each reasoning step should within a set of candidates $\mathcal{S}^t$ to control the computation efficiency. We construct the reasoning step candidates $\mathcal{S}^t$ based on the scoring function defined in Eq. 3 by selecting the top-$m$ reasoning steps with the highest scores. At each timestamp $t$, we leverage the LLMs to select the top-$k$ reasoning steps from the reasoning step candidates $\mathcal{S}^t$. The beam search process at timestamp $t$ is modeled as follows:

$$\mathcal{H}_t = \text{Top}_k \left\{ h \oplus \text{LM}(s^t | q, s^{1:t-1}, w) : h \in \mathcal{H}_{t-1}, s^t \in \mathcal{S}^t \right\} \quad (4)$$

where $\mathcal{H}_{t-1}$ denote the reasoning path up to the previous timestamp $t-1$ and $s^t$ refers to the current reasoning step. $\mathcal{S}^t$ refers to the set of possible next-step candidates retrieved by the Path-RAG at timestamp $t$. $\text{LM}(s^t | q, s^{1:t-1}, w)$ refers to the language models prediction for the next step $s^t$ given the previous sequence $s^{1:t-1}$, the query $q$, and the planning context $w$. The operator $\oplus$ appends $s^t$ to the current path $h$, and $\text{Top}_k$ selects the top-$k$ reasoning paths (with $k$ controlling the beam search width, where a larger $k$ typically yield better performance (see the analysis in Figure 2)).

**Deductive Verification.** To terminate the reasoning path extension process at the right point, we propose to leverage the deductive reasoning capabilities of LLMs (Ling et al., 2024; Huang & Chang, 2023) as a criterion $C(x, s^t, s^{1:t-1}) \in \{0, 1\}$ to automatically guide the decision making of LLMs in Equation 4. Specially, we first leverage LLMs to convert the user query to a declarative statement $q'$ (as shown in Appendix F.6) and then use LLMs to judge whether it can be deduced from current reasoning step $s^t$ and the previous reasoning steps $s^{1:t-1}$. We utilize the same backend LLM with prompt referred in Section F and define the deductive-verification criteria as follows:

$$C(q', s^t, s^{1:t-1}) = \begin{cases} 1, & \text{if } q' \text{ can be deduced from } s^t \text{ and } s^{1:t-1}, \\ 0, & \text{otherwise.} \end{cases} \quad (5)$$

The overall goal of DVBS model can be represented as:

$$\mathcal{H}_t = \text{Top}_B \left\{ h \oplus \text{LM}(s^t | q, s^{1:t-1}, w) : h \in \mathcal{H}_{t-1}, s^t \in \mathcal{S} \text{ and } C(q', s^t, s^{1:t-1}) = 1 \right\} \quad (6)$$

This criterion is an essential component of our method for leveraging LLMs to iteratively validate each step of reasoning, ensuring that each step logically follows from the preceding steps and aligns with the original query. The function effectively signals when to terminate further reasoning, enhancing accuracy and minimizing unnecessary computational efforts.

## 3 Experiments

During the experiment section, we aim to answer the following research questions: **RQ1:** How does FiDeLiS perform compare with existing baselines on KGQA tasks? **RQ2:** How does the design of each individual component contribute to the overall performance of FiDeLiS? **RQ3:** How robustness of FiDeLiS perform under varying conditions? **RQ4:** How about the efficiency of FiDeLiS? **All experiment settings** are detailed in Appendix D due to page constraints.

### 3.1 RQ1: KGQA Performance Comparison

Table 1 clearly showcases the comparative effectiveness of FiDeLiS, against different baselines. It shows that FiDeLiS outperforms all baselines using gpt-4-turbo model, even for strong baselines

Table 1: Comparison of FiDeLiS with baseline methods and different backbone LLMs. We replicate the outcomes of ToG and RoG, and retrieve other baseline results directly from the original paper. We utilize 5 demonstrations as our default setting for FiDeLiS, ToG, Few-shot, and CoT. The experiment results of open-source models can be found in Table 10. In these experiments, we ensured that ToG use the same beam width and depth (set to 4) as FiDeLiS to maintain a fair comparison.

| Backend Models | Methods | WebQSP | | CWQ | | CR-LT |
|---|---|---|---|---|---|---|
| | | Hits@1 (%) | F1 (%) | Hits@1 (%) | F1 (%) | Acc (%) |
| Prompting - LLM Only `gpt-3.5-turbo` | Zero-shot | 54.37 | 52.31 | 34.87 | 28.32 | 32.74 |
| | Few-shot | 56.33 | 53.12 | 38.52 | 33.87 | 36.61 |
| | CoT | 57.42 | 54.72 | 43.21 | 35.85 | 37.42 |
| Prompting - LLM Only `gpt-4-turbo` | Zero-shot | 62.32 | 59.71 | 42.71 | 37.93 | 37.74 |
| | Few-shot | 68.65 | 62.71 | 51.52 | 43.70 | 43.61 |
| | CoT | 72.11 | 65.37 | 53.51 | 44.76 | 45.42 |
| Finetuning - LLM + KG | NSM (He et al., 2021) | 74.31 | - | 53.92 | - | - |
| | CBR-KBQA (Das et al., 2021) | - | - | 67.14 | - | - |
| | DeCAF (Yu et al., 2022) | 82.1 | - | 70.42 | - | - |
| | KD-CoT (Wang et al., 2023a) | 73.7 | 50.2 | 50.5 | - | - |
| | RoG (Luo et al., 2024) | 83.15 | 69.81 | 61.39 | 56.17 | 60.32 |
| Prompting - LLM + KG `gpt-3.5-turbo` | ToG (Sun et al., 2023) | 75.13 | 72.32 | 57.59 | 56.96 | 62.48 |
| | FiDeLiS | 79.32 | 76.78 | 63.12 | 61.78 | 67.34 |
| Prompting - LLM + KG `gpt-4-turbo` | ToG (Sun et al., 2023) | 81.84 | 75.97 | 68.51 | 60.20 | 67.24 |
| | FiDeLiS | **84.39** | **78.32** | **71.47** | **64.32** | **72.12** |

being fine-tuned, such as DeCAF (Yu et al., 2022) and RoG (Luo et al., 2024). Across all prompting-based methods, gpt-4-turbo achieves higher performance compared to gpt-3.5-turbo, especially on the CWQ dataset. It indicates that gpt-4-turbo has a better understanding and processing capability for complex queries. The CR-LT dataset seems more challenging compared with WebQSP and CWQ, as implied by the consistently lower accuracy scores. However, FiDeLiS shows consistent enhancements compared to other baselines, demonstrating its capability to handle long-tail entities by referring to the knowledge from KGs and processing more complex queries effectively.

## 3.2 RQ2: ABLATION STUDY OF FiDeLiS.

Table 2 demonstrates the ablation study of FiDeLiS using the `gpt-3.5-turbo-0125` model across the WebQSP, CWQ, and CR-LT datasets. It highlights the critical importance of both the Path-RAG and DVBS components. Removing Path-RAG, whether by employing a vanilla retriever or the think-on-graph (ToG) approach (Sun et al., 2023), results in substantial performance declines, particularly a 6.97% drop in Hits@1 on WebQSP and a 6.01% decrease on CWQ, underscoring the necessity of an effective retrieval mechanism. Similarly, ablating parts of DVBS, especially the beam search component, leads to significant reductions, with an 18.97% decrease in Hits@1 on WebQSP and a 13.34% drop on CWQ, indicating that beam search is vital for maintaining high accuracy. Other DVBS subcomponents, such as the deductive verifier and planning, also contribute notably to performance, albeit to a lesser degree. Overall, the findings demonstrate that the integrated Path-RAG and DVBS frameworks are essential for FiDeLiS's robust performance, with their removal causing marked decreases in accuracy and Hits@1 scores across all evaluated datasets.

## 3.3 RQ3: ROBUSTNESS ANALYSIS

**Comparison using different embedding backbones.** Table 3 demonstrates the performance of the Path-RAG using various embedding models as backbones. It shows that the integration of the Openai-Embedding Model significantly enhances the performance across all datasets, indicating its superior capability in understanding and processing natural language queries, which leads to more effective retrieval. For instance, the Openai-Embedding Model increases the Hits@1 by 13.04% in the WebQSP dataset and 14.73% in the CWQ dataset compared to the base BM25 (Robertson et al., 2009) model. Furthermore, this improvement is consistent and highlights the feasibility and potential of pairing more powerful language models with structured reasoning frameworks to address complex information retrieval and question answering challenges.

Table 2: Ablation Studies of FiDeLiS using model `gpt-3.5-turbo-0125`. Δ refers to the performance gap between each component and the entire method. The three largest performance gaps on each dataset are highlighted in green, with darker shades denoting more significant differences.

| Ablation Setting | Components | WebQSP | | CWQ | | CR-LT | |
|---|---|---|---|---|---|---|---|
| | | Hits@1 (%) | Δ | Hits@1 (%) | Δ | Acc (%) | Δ |
| No ablation | FiDeLiS | 79.32 | 0.00 | 63.12 | 0.00 | 67.34 | 0.00 |
| w/o Path-RAG | using vanilla retriever | 72.35 | 6.97 | 57.11 | 6.01 | 59.78 | 7.56 |
| | using ToG | 75.11 | 4.21 | 59.47 | 3.65 | 63.47 | 4.07 |
| w/o DVBS | w/o last step reasoning | 75.68 | 3.64 | 59.45 | 3.67 | 63.72 | 3.62 |
| | w/o planning | 76.23 | 3.09 | 60.14 | 2.98 | 64.13 | 3.21 |
| | w/o beam-search | 60.35 | 18.97 | 49.78 | 13.34 | 61.87 | 5.47 |
| | w/o deductive-verifier | 74.13 | 5.19 | 57.23 | 5.89 | 63.89 | 3.45 |

Table 3: Performance FiDeLiS using different Path-RAG embedding backbone models. 'Openai-Embedding Model' refers to `text-embedding-3-small`, the recent embedding model released from OpenAI.

| Methods | Backbones | WebQSP | CWQ | CR-LT |
|---|---|---|---|---|
| | | Hits@1 (%) | Hits@1 (%) | Acc (%) |
| Vanilla Retriever | w/ BM25 (Robertson et al., 2009) | 58.31 | 48.39 | 50.73 |
| | w/ SentenceBert (Reimers & Gurevych, 2019) | 62.74 | 50.14 | 51.80 |
| | w/ E5 (Wang et al., 2022a) | 68.42 | 52.84 | 54.31 |
| | w/ Openai-Embedding-Model | **72.35** | **57.11** | **59.78** |
| Path-RAG | w/ BM25 (Robertson et al., 2009) | 70.34 | 56.11 | 58.77 |
| | w/ SentenceBert (Reimers & Gurevych, 2019) | 73.45 | 58.41 | 60.45 |
| | w/ E5 (Wang et al., 2022a) | 77.93 | 62.74 | 65.23 |
| | w/ Openai-Embedding-Model | **79.32** | **63.12** | **67.34** |

**Comparison using different searching widths and depths.** To investigate how the search width and depth affect the performance of FiDeLiS, we have performed experiments with settings ranging from depths of 1 to 4 and widths of 1 to 4. The results, presented in Figure 2, show that performance generally improves as search depth and width increase. However, performance begins to decline when the search depth exceeds 3 for both the WebQSP and CWQ datasets. This decline is attributed to the fact that only a small fraction of questions in these datasets require reasoning at depths greater than 3. Additionally, FiDeLiS utilizes stricter rules for ending the search process, which can preemptively terminate misguided searches. Contrastingly, increasing the search width consistently shows potential for improved performance, suggesting benefits from broader exploration. However, considering the computational costs, which rise linearly with depth, we have chosen to set both the default beam width and depth to 4 for all experiments unless stated otherwise. This default setting aims to balance performance gains with computational efficiency.

**Comparison between Path-RAG and Vanilla Retriever.** To verify the robustness of the Path-RAG compared to vanilla retrievers. We conduct the experiments as shown in Figure 3a and 3b. We calculate the **coverage ratio (CR)** of the retrieved reasoning paths and the ground-truth reasoning paths as follows: $CR = \left( \frac{N_{\text{retrieved}} \cap N_{\text{ground-truth}}}{N_{\text{ground-truth}}} \right) \times 100\%$, where $N_{\text{retrieved}}$ is the number of retrieved reasoning paths candidates, and $N_{\text{ground-truth}}$ is the total number of ground-truth reasoning paths. We set the vanilla retriever as the baseline, specifically, we concatenate each entity with its corresponding relation to form a reasoning step. We then calculated the cosine similarity between the embeddings of each reasoning steps and the query embeddings to select candidate paths. We observe that compared to the vanilla retriever, Path-RAG achieves a higher CR value and shows better alignment with the ground-truth paths. It demonstrates that our method can better leverage the connections that are often missed by simpler retrieval models. This feature is critical in subsequent LLM to focus on the information of interest and arrive at a correct and reasonable answer.

**Error analysis regarding whole path generation.** To verify the faithfulness of our stepwise method, we conduct the error analysis regarding the whole reasoning path generation. Specifically, we conduct an analysis of the validity of the whole reasoning path generation using the baseline methods RoG (Luo et al., 2024). The result is shown in Figure 3c. The definition of the metric **validity ratio (VR)** is the ratio of reasoning steps that existed in the knowledge graph to the total number of the

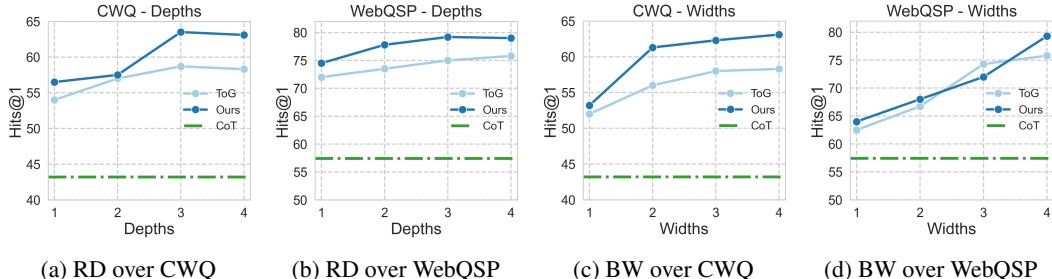

(a) RD over CWQ      (b) RD over WebQSP      (c) BW over CWQ      (d) BW over WebQSP

Figure 2: Performance (Hits@1) of FiDeLiS with different beam search widths (BW) and reasoning depths (RD) over WebQSP and CWQ. (Replaced the figure (d) with the correct figure.) We identified an anomaly in the CWQ data point for ToG with beam-width=4 and beam-depth=4 in Figure 2 (a) and (c). One of the three independent trials for this configuration produced an unusually high score, leading to an inflated average (approximately 60% Hits@1). To address this, we re-ran the experiment under the same configuration and obtained a corrected value of Hits@1 = 58.12%, which falls within the expected variance range observed in similar settings (as reported in Tables 1 and 2). The corrected value has been updated in Figure 2 (a) and (c), and this note has been added to ensure transparency and clarity. This adjustment does not impact any other findings, conclusions, or trends discussed in the paper.

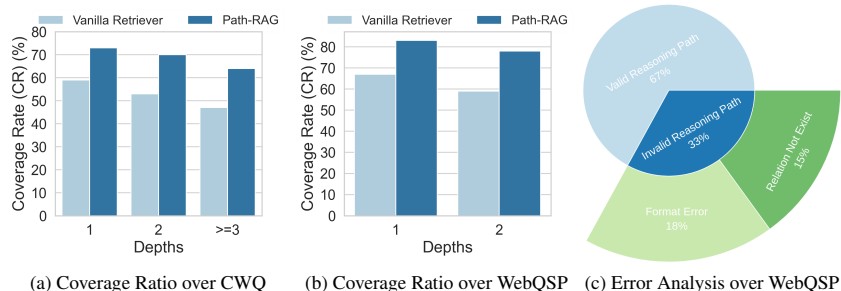

(a) Coverage Ratio over CWQ      (b) Coverage Ratio over WebQSP      (c) Error Analysis over WebQSP

Figure 3: (a)-(b): Empirical study regarding the coverage ratio of the retrieved reasoning paths. (c): Error analysis of the validation of whole paths generated from RoG (Luo et al., 2024) over WebQSP.

reasoning steps in the output reasoning path: $\text{VR} = \left( \frac{N_{\text{valid-steps}}}{N_{\text{all-steps}}} \right) \times 100\%$, where $N_{\text{valid-steps}}$ is the number of reasoning steps that existed in the knowledge graph, and $N_{\text{all-steps}}$ is the number of all the reasoning steps in the reasoning path. It show that only 67% of generated reasoning steps are valid, while the remaining 33% of reasoning steps either have a format error or do not exist in the KG. This illustrates that the reasoning steps generated by language models offer few guarantees about feasibility especially when multiple consecutive steps are combined into a reasoning path.

**Effectiveness of deductive-verification.** To verify the effectiveness of deductive-verification, we calculate the average depths of the grounded reasoning path from ToG and our methods in Table 4. It shows that FiDeLiS consistently shows shorter and closer reasoning depths to ground-truth across all datasets compared to ToG. This implies that our method may offer more precise termination signals and potentially more accurate reasoning paths compared to baselines.

Table 4: Average depths of the returned reasoning paths. GT refers to ground-truth reasoning paths.

| Method | WebQSP | CWQ | CR-LT |
|--------|--------|-----|-------|
| GT | 2.3 | 3.2 | 4.7 |
| ToG | 3.1 | 4.1 | 5.2 |
| FiDeLiS | 2.4 | 2.8 | 4.6 |

**Case Study.** To verify whether FiDeLiS can conduct faithful reasoning on real cases, we also present a case study in Table 5. We can find that FiDeLiS illustrates better performance in identifying multiple possible governmental forms—Theocracy, Unitary state, and Islamic republic established in Iran in 1979, as compared to existing models such as GPT + CoT, RoG, and ToG. This observation highlights the importance of recognizing the complexity of Iran's government system, which combines elements of religion and democracy. Different models handle this complexity with varying levels of detail. For example, GPT + CoT and RoG mainly identified Iran as an 'Islamic Republic', a change that occurred after the 1979 Iranian Revolution, but they missed other possible descriptions. The ToG model even gives mixed results, such as simply naming it 'Iran', which shows confusion in its analysis. However,

our method shows a better understanding of the context and provides a more detailed and accurate set of answers. This illustrates FiDeLiS strength and efficiency in dealing with intricate questions.

Table 5: Case study of FiDeLiS. The wrong answers are highlighted with red color, and the correct answers are highlighted with blue color. In this case, FiDeLiS scored an F1 of 0.857, outperforming the RoG method (0.4).

| Question | What form of government is in the country that uses the Iranian Rail and was established in 1979? |
|---|---|
| Answer | 'Parliamentary system', 'Presidential system', 'Islamic republic', 'Unitary state', 'Theocracy' |
| GPT + CoT | **Think step by step.** First, identify the country: the country that uses the Iranian Rail is Iran. Then, retrieve the historical Context of 1979: In 1979, Iran underwent a significant transformation with the Iranian Revolution, which led to the overthrow of the Pahlavi monarchy. Third, following the revolution, the country adopted a new form of government based on Islamic law. The result was the establishment of the Islamic Republic of Iran, which combines elements of a theocracy and a democracy, where religious leaders have significant influence over the state governance. So, the government form is the Islamic Republic of Iran, established in 1979 following the Iranian Revolution. |
| RoG | location.country.form_of_government, location.location.containedby location.country.form_of_government' location.location.containedby → government.form_of_government.countries
The results are Islamic republic. |
| ToG | **Path-1**: Iranian rial → finance.currency.countries_used → Iran → location.country.form_of_government → Islamic republic → government.form_of_government.countries → Iran
**Path-2**: Iranian rial → finance.currency.countries_used → Iran → location.country.form_of_government → Theocracy → government.form_of_government.countries → Iran
**Path-3**: Iranian rial → finance.currency.countries_used → Iran → location.country.form_of_government → Unitary state → government.form_of_government.countries → Iran
Based on the reasoning paths, the result is Iran. |
| FiDeLiS | **Path-1**: Iranian rial → finance.currency.countries_used → Iran → location.country.form_of_government → Islamic republic
**Path-2**: Iranian rial → finance.currency.countries_used → Iran → location.country.form_of_government → Theocracy
**Path-3**: Iranian rial → finance.currency.countries_used → Iran → location.country.form_of_government → Unitary state
Based on the reasoning paths, the results are Theocracy, Unitary state, Islamic republic. |

## 3.4 RQ4: EFFICIENCY OF FIDELIS

To investigate the runtime efficiency and cost efficiency of FiDeLiS, we present a comparison regarding the average runtime, average token usage, average times of LLM calling per question in Table 6. We found that (1) our method shows superior efficiency compared to the ToG (which is also training-free), by reducing approximately 1.7x runtime costs. (2) Path-Rag component is critical in enhancing both the accuracy and efficiency of the model. Its ability to constrain potential path candidates effectively reduces unnecessary computational overhead, leading to quicker and more accurate results.

Table 6: Runtime efficiency of FiDeLiS per question.

| Dataset | Method | Hits@1 (%) | Avg Runtime (s) | Avg Token Usage | Avg LLM calling |
|---|---|---|---|---|---|
| WebQSP | FiDeLiS (ours) | 79.32 | 43.83 | 2,452 | 10.7 |
| | w/o Path-RAG using vanilla retriever | 72.35 | 48.37 | 2,873 | 10.7 |
| | w/o Path-RAG using ToG | 75.11 | 74.26 | 6,437 | 10.7 |
| | FiDeLiS (ours) - GPT-4o | **81.17** | 37.82 | 2,452 | 10.7 |
| | FiDeLiS (ours) - GPT-4o-mini | 76.48 | **24.31** | 2,452 | 10.7 |
| CWQ | FiDeLiS (ours) | 63.12 | 74.59 | 2,741 | 15.2 |
| | w/o Path-RAG using vanilla retriever | 57.11 | 78.41 | 3,093 | 15.2 |
| | w/o Path-RAG using ToG | 59.47 | 132.59 | 5,372 | 15.2 |
| | FiDeLiS (ours) - GPT-4o | **65.33** | 50.12 | 2,741 | 15.2 |
| | FiDeLiS (ours) - GPT-4o-mini | 58.34 | **42.54** | 2,741 | 15.2 |

To address concern regarding our method's potential application in real-time scenarios, we also tested our method using faster and more advanced LLMs. Table 6 shows that our method could be further accelerated with newer, faster models like GPT-4o or GPT-4-mini. The potential of the ongoing advancements in LLMs are expected to further enhance the scalability and efficiency of FiDeLiS, making it a practical development in challenging environments. More detailed analysis of bottleneck of computation of FiDeLiS can be further found in Appendix C.

## 4 RELATED WORK

**KG-enhanced LLM.** KGs have advantages in dynamic, explicit, and structured knowledge representation and techniques combining LLMs with KGs have been studied (Pan et al., 2024). Early studies (Luo et al., 2024; Yu et al., 2022) embed structured knowledge from KGs into the underlying neural networks during the pretraining or fine-tuning process. However, the reasoning steps generated by LMs are observed to be prone to errors, which could be non-existent in the knowledge graph, and can subsequently lead to incorrect reasoning during inference. Also, KG embedded in LLM sacrifices its own nature of explainability in knowledge reasoning and efficiency in knowledge updating (Hu et al., 2023). Another solutions are to keep the reasoning model as an agent to explore the external structure knowledge source (Sun et al., 2023; Gu et al., 2023; Jiang et al., 2023b; Wang et al., 2023a) ToG (Sun et al., 2023) directly employs LLMs to output scores for candidate selections from the KG. This model operates at the relation-entity level of the KG, aiming to identify relevant triples that aid the LLM in making accurate and responsible final answer predictions.

**Knowledge Graph-based Question Answering.** To integrate LLMs for KGQA, *retrieval-augmented methods* aim to retrieve the relative facts from the KGs to improve the reasoning performance (Li et al., 2023; Karpukhin et al., 2020). Recently, UniKGQA (Jiang et al., 2022) which unifies the graph retrieval and reasoning process into a single model with LLMs, achieves STOA performance. *Semantic parsing methods* convert the question into a structural query (e.g., SPARQL) by LLMs, which can be executed by a query engine to reason the answers on KGs (Sun et al., 2020; Lan & Jiang, 2020). However, these methods heavily rely on the quality of generated queries. If the query is not executable, no answers will be generated. DECAF (Yu et al., 2022) combines semantic parsing and LLMs reasoning to jointly generate answers, which also reach salient performance.

## 5 CONCLUSION

This paper proposes a retrieval-exploration interactive method specifically designed to enhance intermediate steps of LLM reasoning grounded by KGs. The Path-RAG module and the use of deductive reasoning as a calibration tool effectively guide the reasoning process, leading to more accurate knowledge retrieval and prevention of misleading reasoning chains. Extensive experiments demonstrate that our method, being training-free, not only reduces computational costs but also offers superior generality, consistently outperforming established strong baselines across three distinct benchmarks. We believe this study will significantly benefit the integration of LLMs and KGs, or serve as an auxiliary tool to enhance the interpretability and factual reliability of LLM outputs.

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

# A  DEFINITIONS

## A.1  KNOWLEDGE GRAPH-BASED QUESTION ANSWERING (KGQA)

In this work, we mainly focus on the question answering task based on the knowledge graph.

**Knowledge Graphs (KGs)** contain abundant factual knowledge in the form of a set of triples: $\mathcal{G} = \{(e, r, e') \mid e, e' \in \mathcal{E}, r \in \mathcal{R}\}$, where $\mathcal{E}$ and $\mathcal{R}$ denote the set of entities and relations, respectively.

**Knowledge Graph Question Answering (KGQA)** is a typical reasoning task based on KGs. Given a natural language question $q$ and a KG $\mathcal{G}$, the task aims to design a function $f$ to predict answers $a \in \mathcal{A}_q$ based on knowledge from $\mathcal{G}$, i.e., $a = f(q, \mathcal{G})$. Following previous works Luo et al. (2024; 2023), we assume the entities $e_q \in \mathcal{T}_q$ mentioned in $q$ and answers $a \in \mathcal{A}_q$ are labeled and linked to the corresponding entities in $\mathcal{G}$, i.e., $\mathcal{T}_q, \mathcal{A}_q \subseteq \mathcal{E}$.

# B  POTENTIAL IMPACTS AND LIMITATIONS

The proposed method holds the potential to significantly enhance the performance of large language models (LLMs) by tackling the issue of hallucinations, thereby fostering deep, responsible reasoning. By integrating KGs with LLMs, the approach not only facilitates more accurate knowledge retrieval but also leverages deductive reasoning capabilities to steer the reasoning process and circumvent logical fallacies. The method is characterized by its stepwise, generalizable approach, and the use of deductive verification as a stopping criterion, which together may reduce superfluous computation and curb misleading reasoning chains. Moreover, due to its training-free nature and lower computational demands, this method could seamlessly serve as a plug-in for other existing frameworks.

However, it's important to recognize certain limitations. Since the method is still in its early stages of development, it might encounter unanticipated challenges or drawbacks in real-world applications. Its dependency on external KGs means that the quality and comprehensiveness of these resources can affect its overall effectiveness. Computational constraints might also arise when dealing with very large or intricate graphs. Furthermore, although the method has demonstrated promise in benchmark tests, its performance across more diverse or specialized tasks remains untested.

# C  BOTTLENECK OF BEAM SEARCH EFFICIENCY

The bottleneck of computation is the beam search process, which contributes to $N * D$ times LLM calling, where $D$ is the depth (or equivalently length) of the reasoning path, and $N$ is the width of the beam-search (how many paths are remained in the pool in each iteration). Specifically, we need to call $ND + D + C$ times LLM for each sample question, where $C$ is a constant (equals to 1 if there is no error occurs when calling the API). Sun et al. (2023) illustrates that the computational efficiency can be alleviated by replacing LLMs with small models such as BM25 and Sentence-BERT for the beam search decision since the small models are much faster than LLM calling. In this way, we can reduce the number of LLM calling from $ND + D + C$ to $D + C$. However, Sun et al. (2023) shows that this optimization sacrifices the accuracy due to the weaker scoring model in decision making.

We noted that $ND + D + C$ is the maximal computational complexity. In most cases, FiDeLiS does not need $ND + D + C$ LLM calls for a question because the whole reasoning process might be early stopped before the maximum reasoning depth $D$ is reached if LLM determines the query can be deductive reasoning by the current retrieved reasoning paths. As an illustration, Table 6 shows the average numbers of LLM calls per question needed by FiDeLiS on different datasets. It can be seen that in three KGQA datasets, the average numbers of LLM calls (ranging from ) are smaller than 21, which is the theoretical maximum number of LLM calls calculated from $ND + D + C$ when $N = 4$ and $D = 4$. We can also see that this average number gets even smaller for dataset covering a lot of single-hop reasoning questions, such as WebQSP.

# D EXPERIMENT DETAILS

## D.1 BASELINES

**Baselines.** (1) RoG (Luo et al., 2024): embed structure knowledge graph from KGs into the underlying neural networks during the pretraining and fine-tuning process to generate the reasoning path and explanation. (2) ToG (Sun et al., 2023): ask LLM to iteratively explore multiple possible reasoning paths on KGs until the LLM determines that the question can be answered based on the current reasoning paths. Our method FiDeLiS follows the similar paradigm. (3) NSM (He et al., 2021) utilizes the sequential model to mimic the multi-hop reasoning process. (4) KD-CoT (Wang et al., 2023a) retrieves relevant knowledge from KGs to generate faithful reasoning paths for LLMs. (5) DeCAF (Yu et al., 2022) combines semantic parsing and LLMs reasoning to jointly generate answers, which also reach salient performance on KGQA tasks.

**Implementation Details.** We set the default beam width as $4$ and depth as $4$ without specific annotation. We set the $\alpha$ in Eq 3 as 0.3 to ensure reproducibility. For LLMs usage, we set all the inference using temperature $T = 0.3$ and $p = 1.0$. The hyperparameter tuning experiments for beam search and $\alpha$ can be found in Figure 2 and Table 9.

**Backboned LLMs.** We assess our approach on closed- and open-source LLMs. For closed-source LLMs, we choose `GPT-4-turbo`, `GPT-3.5-turbo`[1] to report and compare the results on all datasets. We use `Llama-2-13B` (Touvron et al., 2023) and `Mistral-7B` (Jiang et al., 2023a) as our open-source LLMs to conduct cost–performance analysis on different datasets. The experiment results of open-source models can be found in Table 10.

Table 7: Statistics of the number of answers for questions in WebQSP and CWQ.

| Dataset | #Ans = 1 | $2 \geq$ #Ans $\leq 4$ | $5 \geq$ #Ans $\leq 9$ | #Ans $\geq 10$ |
|---------|----------|------------------------|------------------------|----------------|
| WebQSP  | 51.2%    | 27.4%                  | 8.3%                   | 12.1%          |
| CWQ     | 70.6%    | 19.4%                  | 6%                     | 4%             |

## D.2 DATASETS

**Datasets & Metrics.** We adopt three benchmark KGQA datasets: WebQuestionSP (WebQSP) (Yih et al., 2016), Complex WebQuestions (CWQ) (Talmor & Berant, 2018) and CR-LT-KGQA (Guo et al., 2024) in this work. We follow previous work (Luo et al., 2024) to use the same training and testing splits for fair comparison over WebQSP and CWQ. The questions from both WebQSP and CWQ can be reasoned using Freebase KGs[2]. To address the bias in WebQSP and CWQ, which predominantly feature popular entities and there is a likelihood that their data might have been incorporated into the pre-training corpora of LLMs, we further test our method on CR-LT-KGQA (further discussed in Appendix Section D.2). We use the complete dataset from CR-LT-KGQA in our experiments, as it comprises only 200 samples. Each of the question can be reasoned based on the Wikidata[3]. The statistics of the datasets are given in Table 7 and Table 8. To streamline the KGs, we utilize a subgraph of Freebase by extracting all triples that fall within the maximum reasoning hops from the question entities in WebQSP and CWQ following RoG (Luo et al., 2024). Similarly, we construct the corresponding sub-graphs of Wikidata for CR-LT-KGQA as well. We assess the performance of the methods by analyzing the F1 and Hits@1 metrics for CWQ and WebQSP, and by evaluating the accuracy for CR-LT-KGQA.

**Motivation of CR-LT-KGQA.** The motivation for evaluating over CR-LT-KGQA is that the majority of existing KGQA datasets, including WebQSP and CWQ, predominantly feature popular entities. These entities are well-represented in the training corpora of LLMs, allowing to often generate correct answers based on their internal knowledge, potentially without external KGs. Moreover, since WebQSP and CWQ have been available for several years, there is a likelihood that their data might

---

[1]We use the recent `gpt-4-turbo` model that is released in 2024-04-09 from https://platform.openai.com/docs/models/gpt-4-turbo-and-gpt-4 and `gpt-3.5-turbo` model notated as "gpt-3.5-turbo-0125" from https://platform.openai.com/docs/models/gpt-3-5-turbo.

[2]https://github.com/microsoft/FastRDFStore

[3]https://www.wikidata.org/wiki/Wikidata:Main_Page

Table 8: Statistics of the question hops in WebQSP, CWQ and CR-LT-KGQA.

| Dataset | 1 hop | 2 hop | $\geq$ 3 hop |
|---------|-------|-------|--------------|
| WebQSP  | 65.49 % | 34.51% | 0.00% |
| CWQ     | 40.91 % | 38.34% | 20.75% |
| CR-LT   | 5.31 %  | 43.22% | 51.57% |

have been incorporated into the pre-training corpora of LLMs, further reducing the need for external KGs during question-answering.

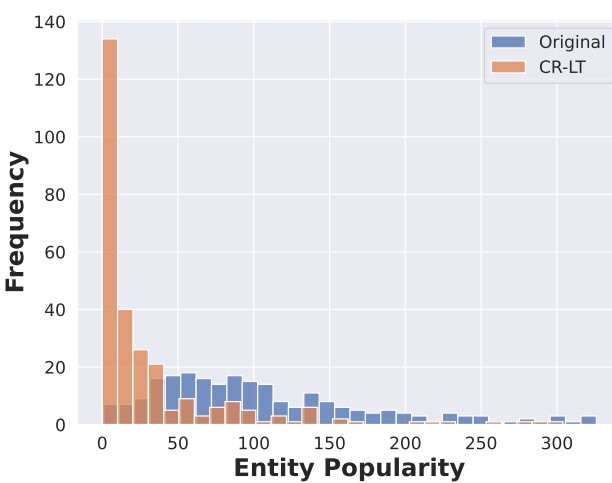

Figure 4: Distribution of CR-LT-KGQA dataset.

Against this backdrop, we specifically chose CR-LT-KGQA for our evaluation as it includes queries related to obscure or long-tail entities, the distribution of the frequency and entity popularity of CR-LT is shown as Figure 4. Such scenarios are where KGs play a critical role because they provide a reliable source of verifiable information, crucial when LLMs encounter entities not well covered in their training data. By testing our methods on CR-LT-KGQA, we aim to explore how effective LLMs can operate when combined with KGs, especially in contexts involving less common knowledge domains where LLM performance typically declines. This evaluation helps us understand the extent to which KGs remain necessary for supporting LLMs in a diverse range of query scenarios.

### D.3 PARAMETER TUNING FOR $\alpha$ FOR SCORING FUNCTION

For hyperparameter tuning regarding $\alpha$ for Eq 3, we added an extra comparison in Table 9 which shows that $\alpha$ is actually not a very impact parameter for the whole system, however for the reproducibility of our method, we set the $\alpha$ as 0.3.

Table 9: Parameter tuning for $\alpha$ for scoring function over WebQSP

| $\alpha$ | 0.1 | 0.2 | 0.3 | 0.4 | 0.5 | 0.6 | 0.7 | 0.8 | 0.9 | 1.0 |
|----------|-----|-----|-----|-----|-----|-----|-----|-----|-----|-----|
| Hits@1 | 83.76 | 84.15 | 84.39 | 83.98 | 83.55 | 84.12 | 83.79 | 82.25 | 83.60 | 83.73 |

# E ALGORITHM DESCRIPTION

---

**Algorithm 1** Path-RAG Initialization and Retrieval Process

---

1: **Initialization:**
2: **for all** $e_i \in \mathcal{E}, r_i \in \mathcal{R}$ **do**
3: $\quad z_e^i = \text{LM}(e_i)$          ▷ Embed entities
4: $\quad z_r^i = \text{LM}(r_i)$          ▷ Embed relations
5: **end for**
6: Populate nearest neighbor index with $\{z_e^i\}$ and $\{z_r^i\}$      ▷ Facilitate retrieval
7: **procedure** RETRIEVE(query $q$)
8: $\quad \mathcal{K}_i = \text{LM}(\text{'prompt'}, q)$          ▷ Generate keywords
9: $\quad$ **for all** $k_i^m \in \mathcal{K}_i$ **do**
10: $\quad\quad k_i \leftarrow \text{concatenate}(k_i^m)$
11: $\quad\quad z_k = \text{LM}(k_i)$          ▷ Embed concatenated keywords
12: $\quad\quad \mathcal{E}_k = \text{argtopk}_{i \in \mathcal{E}} \cos(z_k, z_e^i)$      ▷ Retrieve top-k entities
13: $\quad\quad \mathcal{R}_k = \text{argtopk}_{i \in \mathcal{R}} \cos(z_k, z_r^i)$      ▷ Retrieve top-k relations
14: $\quad$ **end for**
15: $\quad$ **return** $\mathcal{E}_k, \mathcal{R}_k$
16: **end procedure**
17: **procedure** SCOREPATH($\mathcal{E}_k, \mathcal{R}_k$)
18: $\quad$ Initialize Score $\leftarrow 0$
19: $\quad$ **for** each $e_k \in \mathcal{E}_k$ and $r_k \in \mathcal{R}_k$ **do**
20: $\quad\quad$ Calculate $S_e^i, S_r^i \leftarrow \cos(z_k, z_e^i), \cos(z_k, z_r^i)$      ▷ Compute similarity scores
21: $\quad\quad S(p) = S_r^i + S_e^i + \alpha \max_{\forall j \in N_i}(S_r^j + S_e^j)$      ▷ Score path using Eq. 3
22: $\quad\quad$ Score $\leftarrow \max(\text{Score}, S(p))$      ▷ Update max score
23: $\quad$ **end for**
24: $\quad$ **return** Score, $p$
25: **end procedure**

---

**Algorithm 2** Deductive-Verification Guided Beam Search

---

**Require:** User query $x$, Beam width $B$
**Ensure:** Reasoning path $s^{1:T}$
1: Initialize $\mathcal{H}_0 = \{\emptyset\}$
2: Utilize LLM to generate from $x$:
3: $\quad$ Planning steps.
4: $\quad$ Declarative statement $x'$.
5: **for** $t = 1$ to $T$ **do**
6: $\quad$ **for** each $h \in \mathcal{H}_{t-1}$ **do**
7: $\quad\quad$ Generate possible next steps $s^t \in \mathcal{S}$ using Path-RAG.
8: $\quad\quad$ **for** each $s^t$ **do**
9: $\quad\quad\quad$ Compute $C(x', s^t, s^{1:t-1})$ using LLM:
10: $$C(x', s^t, s^{1:t-1}) = \begin{cases} 1 & \text{if } x' \text{ can be deduced from } s^t \text{ and } s^{1:t-1}, \\ 0 & \text{otherwise.} \end{cases}$$
11: $\quad\quad\quad$ **if** $C(x', s^t, s^{1:t-1}) = 1$ **then**
12: $\quad\quad\quad\quad$ Append $s^t$ to $h$ to form new hypothesis $h'$.
13: $\quad\quad\quad\quad$ Add $h'$ to $\mathcal{H}_t$.
14: $\quad\quad\quad$ **end if**
15: $\quad\quad$ **end for**
16: $\quad$ **end for**
17: $\quad$ $\mathcal{H}_t = \text{Top}_B(\mathcal{H}_t)$ based on scoring function (like plausibility or likelihood).
18: **end for**
19: **return** the best hypothesis from $\mathcal{H}_T$.

---

# F  PROMPT LIST

In this section, we show all the prompts that need to be used in the main experiments. The `In-Context Few-shot` refers to the few-shot examples we used for in-context learning.

## F.1  PLAN-AND-SOLVE

You are a helpful assistant designed to output JSON that aids in navigating a knowledge graph to answer a provided question. The response should include the following keys:

(1) 'keywords': an exhaustive list of keywords or relation names that you would use to find the reasoning path from the knowledge graph to answer the question. Aim for maximum coverage to ensure no potential reasoning paths will be overlooked;

(2) 'planning_steps': a list of detailed steps required to trace the reasoning path with. Each step should be a string instead of a dict.

(3) 'declarative_statement': a string of declarative statement that can be transformed from the given query, For example, convert the question 'What do Jamaican people speak?' into the statement 'Jamaican people speak *placeholder*.' leave the *placeholder* unchanged; Ensure the JSON object clearly separates these components.

`In-Context Few-shot`

Q: {Query}

A:

## F.2  DEDUCTIVE-VERIFICATION

You are asked to verify whether the reasoning step follows deductively from the question and the current reasoning path in a deductive manner. If yes return yes, if no, return no".

`In-Context Few-shot`

Whether the conclusion '{declarative_statement}' can be deduced from '{parsed_reasoning_path}', if yes, return yes, if no, return no.

A:

## F.3  ADEQUACY-VERIFICATION

You are asked to verify whether it's sufficient for you to answer the question with the following reasoning path. For each reasoning path, respond with 'Yes' if it is sufficient, and 'No' if it is not. Your response should be either 'Yes' or 'No'.

`In-Context Few-shot`

Whether the reasoning path '{reasoning_path}' be sufficient to answer the query '{Query}', if yes, return yes, if no, return no.

A:

## F.4  BEAM SEARCH

Given a question and the starting entity from a knowledge graph, you are asked to retrieve reasoning paths from the given reasoning paths that are useful for answering the question.

`In-Context Few-shot`

Considering the planning context {plan_context} and the given question {Query}, you are asked to choose the best {beam_width} reasoning paths from the following candidates with the highest probability to lead to a useful reasoning path for answering the question. {reasoning_paths}. Only return the index of the {beam_width} selected reasoning paths in a list.

A:

## F.5 REASONING

Given a question and the associated retrieved reasoning path from a knowledge graph, you are asked to answer the following question based on the reasoning path and your knowledge. Only return the answer to the question.

```
In-Context Few-shot
```

Question: {Query}

Reasoning path: {reasoning_path}

Only return the answer to the question.

A:

## F.6 DEMONSTRATION OF DEDUCTIVE VERIFICATION

---

**Deductive Verification Example**

**Question:** Who is the ex-wife of Justin Bieber's father?

After one round of beam searching, the **current reasoning path** is:
*Justin_bieber → people.person.father → Jeremy_bieber.*

The **next step candidates** are:
1. *people.married_to.person → Erin Wagner*
2. *people.person.place_of_birth → US, . . .*

The deductive reasoning can be formulated as follows:

**Premises:**

- Justin_bieber → people.person.father → Jeremy_bieber
(from the current reasoning path)
- Jeremy_bieber → people.married_to.person → Erin Wagner
(from the next step candidates)

**Conclusion:**

Erin Wagner is the ex-wife of Justin Bieber's father.
(Using a large language model (LLM) zero-shot approach to reformat the question into a cloze filling task, we use the last entity from the next step candidates, "Erin Wagner", to fill the cloze.)

The prompt will ask whether the conclusion can be deduced from the given premises. If the answer is "yes", return "yes", otherwise return "no."

---

## G RELATED WORKS: REASONING WITH LLM PROMPTING

With the development of LLMs, many creative ways to leverage LLMs are proposed: End-to-End, Chain-of-Thought, and Semantic Parsing/Code Generation.

Table 10: Comparison of FiDeLiS using different backbone LLMs.

| Backend Models | Methods | WebQSP | | CWQ | | CR-LT |
|---|---|---|---|---|---|---|
| | | Hits@1 (%) | F1 (%) | Hits@1 (%) | F1 (%) | Acc (%) |
| Llama-2-13B | FiDeLiS | 72.34 | 69.78 | 58.41 | 54.78 | 60.87 |
| Mistral-7B | FiDeLiS | 74.11 | 70.23 | 60.71 | 56.87 | 63.12 |

**End-to-End** methods (Chen, 2022; Hoffmann et al., 2022) aims to leverage LLMs to generate final answers directly, often done by providing a task description and/or a few examples for in-context learning. While it offers convenience, however, it suffers from the un-interpretability of the generation, and the lack of explicit steps and reliance solely on the LLMs' training data may result in un-robustness and exhibit sensitivity to slight input variations.

**Chain-of-Thought** methods (Wei et al., 2022a; Kojima et al., 2022; Gong et al., 2020) emphasize breaking down complex reasoning into a series of intermediate steps to support complex reasoning. However, CoT suffers from unreliability and uncontrollability since the generated reasoning steps may not always align with the intended logical progression or may produce incorrect or inconsistent answers when multiple questions are posed consecutively.

**Plan-and-solve** methods (Wang et al., 2023b) prompts LLMs to generate a plan and conduct reasoning based on it. DecomP He et al. (2021) prompts LLMs to decompose the reasoning task into a series of sub-tasks and solve them step by step.

**Semantic Parsing/Code Generation** methods (Cheng et al., 2023; Ye et al., 2023; Gemmell & Dalton, 2023) leverages LLMs to convert natural language queries into executable code or structured representations. This approach enables more precise control over the output and facilitates better interpretability. However, it has its limitations, such as the limited coverage of programming language grammar and semantics. This restricts the model's ability to handle complex programming tasks and may require additional techniques to handle out-of-domain queries effectively. Cheng et al. (2023) propose to leverage LLMs to bridge the out-of-domain queries, however, the form of knowledge injection still lacks a thorough exploration and the trigger functions are still naive for complex queries. However, the problem of hallucinations and lack of knowledge affect the faithfulness of LLMs' reasoning. ReACT Yao et al. (2022) treats LLMs as agents, which interact with the environment to get the latest knowledge for reasoning. To explore faithful reasoning, FAME Hong et al. (2023) introduces the Monte-Carlo planning to generate faithful reasoning steps. RR He et al. (2022) and KD-CoT Wang et al. (2023a) further retrieve relevant knowledge from KGs to produce faithful reasoning plans for LLMs.

## H  ROBUSTNESS ANALYSIS ACROSS DIFFERENT KG PERTURBATION

To further mimic the real situations where KGs may not be of high quality (*i.e.*, attributes of nodes/edges may be mislabeled, relations may not exist, *etc.*), we propose another experiment setting in this section to assess the model performance under conditions where KGs' semantics and structure are deliberately perturbed and contaminated. Considering that KGs are typically annotated by humans and are generally accurate and meaningful, we introduce perturbations to edges in the KG to degrade the quality of the KGs.

For the perturbation methods, we consider four perturbation heuristics based on (Raman et al., 2020) as follows: **Relation Swapping (RS)** randomly chooses two edges from $\mathcal{T}$ and swaps their relations. **Relation Replacement (RR)** randomly chooses an edge $v_1, e, v_2) \in \mathcal{T}$, then replaces $e_1$ with another relation $e_2 = \text{argmin}_{r \in \mathcal{R}} S_{\mathcal{G}}(v_1, e, v_2)$, where $S_{\mathcal{G}}(v_1, e, v_2)$ uses ATS to measure the semantics similarity between two edges. **Edge Rewiring (ER)** randomly chooses an edge $(v_1, e, v_2) \in \mathcal{T}$, then replaces $v_2$ with another entity $v_3 \in \mathcal{E} \backslash \mathcal{N}_1(v_1)$, where $\mathcal{N}_1(v_1)$ represents the 1-hop neighborhood of $v_1$. **Edge Deletion (ED)** randomly chooses an edge $(v_1, e, v_2) \in \mathcal{T}$ and deletes it. We control the perturbation level based on the percentage of KG edges being perturbed.

Figure 5 indicates that while the performance of our method does degrade under such conditions, it remains robust to a reasonable level of noise. This robustness is primarily due to our method's reliance on both semantic similarity and structural information during retrieval, which helps mitigate

the effects of incorrect or incomplete edges. Additionally, the LLM's reasoning capabilities provide further resilience by dynamically compensating for some inaccuracies in the retrieved reasoning paths.

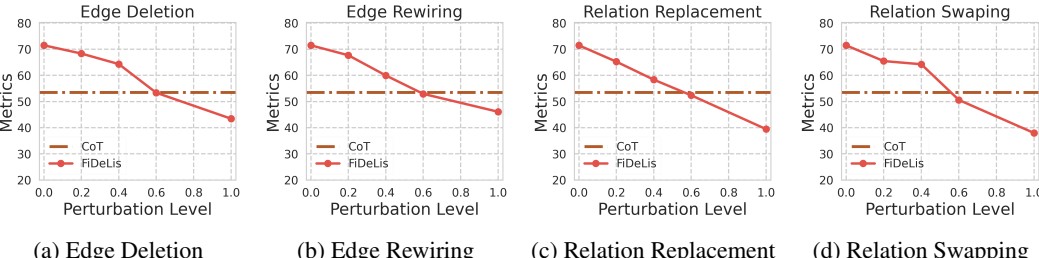

(a) Edge Deletion   (b) Edge Rewiring   (c) Relation Replacement   (d) Relation Swaping

Figure 5: Performance Metric (His@1) vs. Perturbation Level for Different Perturbation Methods and Different Retrieval Methods.

