# OpenReview forum: "FiDeLiS: Faithful Reasoning in Large Language Model for Knowledge Graph Question Answering"
_ICLR.cc/2025/Conference — Submitted to ICLR 2025_

### Official Review · Reviewer_rRZc · 2024-10-31

**Soundness:** 3
**Presentation:** 3
**Contribution:** 3
**Rating:** 6
**Confidence:** 4

**Summary:**

This paper proposes a method for extracting knowledge from a Knowledge Graph to enhance model reasoning capabilities. The core innovation of this paper lies in the Deductive-Verification Guided Beam Search, which enhances efficiency by allowing the model to select the top-k reasoning steps and implement pruning during path extension.

**Strengths:**

1. The application of this approach demonstrated significant improvements across various metrics on three KGQA datasets.
2. This study substantiates that knowledge graphs can to a certain degree mitigate the hallucination problem in model reasoning within open-domain question answering.

**Weaknesses:**

1. Several sections of the manuscript do not adequately convey essential information. For example, the title lacks precision in capturing the central elements of the study, and the abstract does not sufficiently highlight the primary research questions or issues.
2. In the experiments, comparing Fine-tuned LLM with GPT-4 seems unfair.
3. The Vanilla retriever compared in Section 3.3 is a relatively simple retriever.

**Questions:**

In retrieving specific knowledge from a knowledge graph, this paper proposes a method based on keyword extraction from the question and similarity assessment. However, Section 3.3 only compares embedding models. Does the accuracy of keyword extraction significantly impact the effectiveness of the retrieval approach?

---

> ### Author Response · Authors · 2024-11-23
>
> We sincerely thank the reviewer for their thoughtful comments. Regarding the writing issues mentioned in **W1**, we will polish our paper in the revised version. For the other concerns, we provide detailed responses below:
>
> ---
> ### **W2: The experiments comparing Fine-tuned LLMs with GPT-4 seems unfair.**
>
> We acknowledge the concern due to GPT-4 being a proprietary model with limited accessibility for fine-tuning or deeper customization. Our decision for this comparison is intended to observe trends whether the proposed **training-free framework** can achieve similar or even better performance compared to methods requiring further training. Our findings demonstrate that the proposed method, as a training-free framework, outperforms fine-tuned models while requiring significantly less time and computational resources for training.
>
> ---
> ### **W3: The vanilla retriever compared in Section 3.3 is a relatively simple retriever.**
>
> We appreciate the reviewer's concern. To further validate this, we add another baseline from KAPING[1], and report the coverage ratio comparison on CWQ as follows:
>
> | Method | Depth=1 | Depth=2 | Depth>3 |
> |---|:---:|:---:|:---:|
> | Vanilla Retriever | 59.34 | 52.17 | 47.31 |
> | KAPING (top-k triplet retrieval) | 65.72 | 60.41 | 53.11 |
> | Path-RAG | 72.61 | 69.38 | 62.78 |
>
> These results demonstrate that Path-RAG consistently achieves higher coverage, especially for deeper reasoning paths. Unlike the triplet-based retrieval in KAPING, Path-RAG leverages graph structure to capture not only highly relevant nodes and edges but also intermediate “bridge” connecting other highly relevant nodes and edges in next hop neighbors. This design mitigates over-reliance on semantic similarity alone by ensuring that structural relationships are also considered, reducing the likelihood of errors in reasoning path construction.
>
> * [1] Knowledge-Augmented Language Model Prompting for Zero-Shot Knowledge Graph Question Answering: https://arxiv.org/pdf/2306.04136
>
> ---
> ### **Q1: Does the accuracy of keyword extraction significantly impact the effectiveness of the retrieval approach?**
>
> Yes, the keyword extraction impacts the effectiveness of the retrieval approach. We add another ablation study to consider using keywords and without using keywords as follows: (we use coverage ratio to measure the effectiveness of different retrievers)
>
> | Method | Depth=1 | Depth=2 | Depth>3 |
> |---|:---:|:---:|:---:|
> | Path-RAG w/ keywords | 72.61 | 69.38 | 62.78 |
> | Path-RAG w/o keywords  | 68.78 (**$\downarrow$ 3.83**) | 65.27 (**$\downarrow$ 4.11**) | 57.13 (**$\downarrow$ 5.65**) |
>
> It shows that generating an exhaustive list of keywords related to the query can maximize the coverage of potential reasoning steps required to answer it. These keywords expand the search space for the beam search process by including additional, related information that may not be explicitly stated in the query. During the retrieval stage, this broader keyword list allows the system to identify more potential candidates, which enriches the input for the beam search. As a result, the expanded search space increases the chances of discovering relevant reasoning paths and improves the model’s ability to find accurate and effective solutions.

---

> > ### Comment · Reviewer_rRZc · 2024-11-27
> >
> > Thank you for your response. Based on the additional keyword extraction ablation experiments, it appears that the role of keywords in your approach is minimal, with only a 3.83% improvement at depth 1. However, this raises another point of confusion for me—if the keyword extraction step is completely removed, how do the subsequent steps of the approach function? My original intention was to evaluate how the correctness of keyword extraction impacts the final results. Overall, while I think there is value to the theoretical findings of the paper, I keep my score.

---

> > > ### Author Response · Authors · 2024-11-27
> > >
> > > Thanks for the follow-up question. We would like to clarify that the setting w/o keywords refers to when we only use the original query instead of the keywords to calculate the similarity score as shown in Eq (2), and for the subsequent processing, we keep them the same to make the comparison fair.
> > >
> > > **w/ keywords** (where $K$ refers to the keywords):
> > >
> > > $E_m = \operatorname{arg\,top_m}_{i \in E} \cos(z(K), z(e))$
> > >
> > > $R_m = \operatorname{arg\,topm}_{i \in R} \cos(z(K), z(r))$
> > >
> > > **w/o keywords** (where $q$ refers to the user query):
> > >
> > > $E_m = \operatorname{arg\,top_m}_{i \in E} \cos(z(q), z(e))$
> > >
> > > $R_m = \operatorname{arg\,topm}_{i \in R} \cos(z(q), z(r))$
> > >
> > > In addition, we would like to highlight that this improvement on the retrieval stage should not be considered minimal considering the large scale of the knowledge graph, where even small gains reflect improvements in the system's ability to retrieve relevant reasoning steps amidst vast amounts of candidates. Additionally, the benefit of adjusting keywords is more notable in queries that require deeper reasoning, with improvements of 4.11% at depth 2 and 5.65% at depths greater than 3. It's also important to note that keyword utilization is only one aspect of our approach. Our method outperforms standard baseline retrievers like KAPING, achieving even more substantial performance enhancements (as shown in the above response).
> > >
> > > We are eager to address any additional concerns you might have. Please let us know if there are specific issues or expectations we should meet to improve our rating. We are committed to responding effectively to your feedback. Thanks.

---

### Official Review · Reviewer_mHFm · 2024-10-31

**Soundness:** 3
**Presentation:** 2
**Contribution:** 2
**Rating:** 5
**Confidence:** 3

**Summary:**

This paper introduces a novel method combining LLMs and KGs, primarily consisting of two modules: Path-RAG and DVBS. Specifically, Path-RAG is responsible for retrieving relevant information from knowledge graphs, while DVBS selects the most promising reasoning paths through beam search. I am pleased to see the experimental performance across three KGQA datasets, where the authors' method shows improvements in both accuracy and efficiency. Additionally, the authors conducted extensive ablation experiments, which enhances the paper's soundness.

**Strengths:**

- The authors' writing is clear, with well-explained methodology.
- The authors achieved state-of-the-art performance across multiple KGQA datasets, demonstrating consistent performance improvements.

**Weaknesses:**

(The following weaknesses represent my second version, which incorporates feedback from the Associate Program Chairs)

- The paper's primary contribution appears incremental, as it primarily combines existing retrieval and ranking mechanisms without introducing fundamentally new theoretical insights or technical innovations - the authors should clarify what specific technical advances differentiate their approach from previous retrieval-augmented systems.

- Path-RAG appears to be a complex retrieval mechanism, and given that most entities and relations are connected in the knowledge graph, it's unclear how this approach is beneficial. In my view, Path-RAG might be more useful in cases where entities and relations are not directly connected. However, the paper doesn't address how the system handles multi-hop reasoning chains. Additionally, the scoring function is influenced by $alpha$, but it's unclear how $alpha$ is determined. Only qualitative results seem to be provided.

- The paper only compares with GoT, missing some new baseline methods, including KG-CoT[1], ToG2[2], and GNN-RAG[3]. the authors should either include these comparisons or provide compelling justification for their exclusion.

- Furthermore, there are areas for experimental improvement. The experimental methodology would benefit from evaluation on more recent language models (such as open-source alternatives or current SOTA models) to demonstrate the robustness and generalizability of the proposed approach across different model architectures.



---

[1]  KG-CoT: Chain-of-Thought Prompting of Large Language Models over Knowledge Graphs for Knowledge-Aware Question Answering

[2] Think-on-Graph 2.0: Deep and Faithful Large Language Model Reasoning with Knowledge-guided Retrieval Augmented Generation

[3]  GNN-RAG: Graph Neural Retrieval for Large Language Model Reasoning

**Questions:**

Can you provide a justification or explanation for the concept of "deductive verification"? I have several concerns regarding the proposed method's ability to perform deductive verification, which is a significant claim within the paper. The model appears to lack a structured knowledge base or set of rules from which it can draw conclusions, which would be necessary for true deductive verification. Given this, I am skeptical of the term "deductive verification" being used in this context.

---

> ### Author Response · Authors · 2024-11-23
>
> We sincerely thank the reviewer for the thoughtful comments. Below, we address each concern in detail:
>
> ---
> ### **W1: The reviewer is concerned that the novelty of the paper is limited, especially compared with existing retrieval-augmented methods.**
>
> We acknowledge that there are similarities between the proposed method and previous retrieval-augmented methods, especially we both follow the workflow of retrieving some information from external resource and use them to enhance the LLM reasoning. However, we would like to highlight that our contributions should be not considered as incremental efforts for the following reasons:
>
> **Paradigm shift from retrieving knowledge facts to reasoning paths from KG.** Unlike existing retrieval-augmented methods that retrieve individual triplets from a knowledge graph (KG) as additional knowledge to support reasoning, our method leverages the structural information of the KG. Specifically, we retrieve reasoning paths—sequences of interconnected triplets—directly from the KG to guide LLM reasoning. These reasoning paths provide a more structured and factual basis for reasoning compared to the self-generated chain-of-thought (CoT) by LLMs, ensuring greater factual accuracy. Additionally, the reasoning paths enhance the explainability of the reasoning process, offering a clear rationale for how the final answer is derived.
>
> In addition, our proposed method focus on **two challenges** when retrieving reasoning paths from KG as follows:
>
> **(a) Issues of premature stopping or excessive continuation when extending the reasoning paths**: When retrieving reasoning paths from a KG, two critical challenges can arise: premature stopping, where the retrieval process halts before a complete reasoning path is constructed, or excessive continuation, where the path is extended unnecessarily, including irrelevant or incorrect steps. These issues will hinder the performance of LLMs’ decision making (as shown in the case study from lines 422 to 455). Our proposed method uses deductive reasoning as a clear and objective way to decide when to stop extending reasoning paths. Deductive reasoning involves verifying whether each reasoning step logically follows from the previous steps and the user query, making it more reliable and less ambiguous. This approach not only simplifies decision-making but also reduces bias in the reasoning process, ensuring fairer and more accurate stopping criteria. We would like to highlight that this straightforward shift in controlling the end point of reasoning paths demonstrates promising performance, particularly in cases requiring deeper reasoning steps (e.g., CoT and CL-RT with reasoning depths >3).
>
> **(b) Issues of inefficiency in handling large reasoning steps candidates**: In addition, previous work did not consider the retrieval process, which means it considers all neighboring entities/relations during reasoning path extension. This can be particularly problematic for large KGs that have a vast number of neighbors available. It will substantially increase the computational cost for LLM reasoning (as more tokens are considered) and introduces noise from irrelevant nodes and edges, which hampers the effectiveness of subsequent LLM’s decision making process. In contrast, our method first retrieves entities and relations relevant to the query to construct reasoning path candidates. This allows the LLM to focus on a **smaller, more relevant set of candidates** during decision-making, improving efficiency and reducing the need to filter out irrelevant noise. As demonstrated in our experiments, this retrieval-based approach (path-RAG) consistently achieves better overall performance on different settings (Table 1,2).
>
> ---
> ### **W2: Why use Path-RAG and what is the sensitivity of the hyperparameter $\alpha$?**
>
> Path-RAG is proposed to retrieve entities and relations relevant to the query to construct reasoning path candidates. This allows the LLM to focus on a smaller, more relevant set of candidates during decision-making, improving efficiency and reducing the need to filter out irrelevant noise. As demonstrated in our experiments, this retrieval-based approach (path-RAG) consistently achieves better overall performance on different settings (Table 1,2).
>
> Quantitatively, the hyperparameter $\alpha$ in the scoring function (Eq. (3)) balances short-term outcomes and long-term potential in reasoning paths. A higher $\alpha$ prioritizes paths with long-term benefits, even if they appear sub-optimal initially, whereas a lower $\alpha$ emphasizes immediate gains, potentially overlooking future impacts. We select the $\alpha$ using grid search. As noted in lines 849 to 855, $\alpha$ does not significantly affect the overall system performance, indicating that Eq. (3) is not highly sensitive to variations in $\alpha$.

---

> ### Author Response · Authors · 2024-11-23
>
> ---
> ### **W3: Adding more recent baselines and more model backbones.**
>
> We acknowledge that **KG-CoT** and **ToG-2.0** were recently published and are contemporaneous with our own. As they became available around the same time as our submission, we have not considered them as primary baselines in our study. However, we appreciate the reviewers bringing these to our attention and will take them into account in future work.
>
> For the reference **GNN-RAG**, we acknowledge that this paper have shown better performance as a framework requires extra training, while our proposed method is more like a training-free framework. The comparison between these two frames may not be very fair.
>
> To address the reviewers’ concerns about generalizability and robustness across different model architectures, we add more recent language models as follows:
>
> | Methods | WebQSP (hits@1) | CWQ (hits@1) |
> |---|:---:|:---:|
> | FiDeLiS+gpt-4o | 86.34 | 73.48 |
> | FiDeLiS+qwen-2-7B | 64.32 | 50.79 |
> | FiDeLiS+llama-3.1-8B | 74.41 | 55.73 |
> | FiDeLiS+mixtral-8x7b-instruct | 68.13 | 52.37 |
>
> ---
> ### **Q1: Explanation for the concept of "deductive verification"**
>
> Thank you for pointing out this concern. We understand that the term “deductive verification” might traditionally suggest the use of a formal, rule-based system or a structured knowledge base. In our work, **however**, the term is used more broadly to describe how the LLM infers logical consequences by evaluating whether a given reasoning path aligns with the user query and retrieved evidence from the knowledge graph.
>
> For instance, as described in the manuscript (Lines 989–1020), consider the question: *“Who is the ex-wife of Justin Bieber’s father?”* After one round of beam searching, the current reasoning path is:
>
> **“Justin_bieber → people.person.father → Jeremy_bieber.”**
>
> The next step candidates are:
> 1. *people.married_to.person → Erin Wagner*
> 2. *people.person.place_of_birth → US*, . . .
>
> In this context, **“deductive reasoning verification”** operates as follows:
> The model evaluates whether the user query can be logically deduced by extending the current reasoning path with a candidate step. For instance, we assess whether the candidate **people.married_to.person → Erin Wagner** logically supports the query. Specifically, we represent reasoning paths as premises and the user query as a conclusion, then check whether the conclusion can be deduced from the premises.
>
> **Premise**:
> - Justin\_bieber $\to$ people.person.father $\to$ Jeremy\_bieber (**from the current reasoning path**)
> - Jeremy\_bieber $\to$ people.married\_to.person $\to$ Erin Wagner (**from the next step candidates**)
>
> **Conclusion**:
> - Erin Wagner is the ex-wife of Justin Bieber’s father.
> (Using a large language model (LLM) zero-shot approach to reformat the question into a cloze filling task, we use the last entity from the next step candidates, "Erin Wagner", to fill the cloze.)
>
> The LLM is prompted to evaluate whether the conclusion logically follows from the premises. If the answer is “yes,” the reasoning path extension is considered complete. If the answer is “no,” the reasoning path is either extended further or discarded entirely.
> This approach differs from traditional rule-based systems as it does not rely on a predefined set of rules or formalized knowledge bases. Instead, it leverages the LLM’s implicit knowledge and reasoning capabilities to dynamically evaluate each step in the context of the query

---

> > ### Comment · Reviewer_mHFm · 2024-11-25
> > **Thans for your response!**
> >
> > Thank you to the author for addressing most of my concerns. However, I remain quite concerned about the incremental nature of the method's improvements. Especially after seeing your response that "Paradigm shift from retrieving knowledge facts to reasoning paths from KG," in my view this approach represents an incremental improvement on existing work. Considering that many previous works have already explored using reasoning paths from KG to enhance RAG and KBQA tasks (dating back to the BERT era) [1,2,3], I believe the technical contribution of this paper may not meet ICLR's bar. Therefore, I will maintain my current score for now, though I remain optimistic and am eager to discuss this work's contributions with other reviewers!
> >
> > ---
> >
> > [1] Lan Y, He S, Liu K, et al. Path-based knowledge reasoning with textual semantic information for medical knowledge graph completion[J]. BMC medical informatics and decision making, 2021, 21: 1-12.
> >
> > [2] Li Z, Jin X, Guan S, et al. Path reasoning over knowledge graph: A multi-agent and reinforcement learning based method[C]//2018 IEEE International Conference on Data Mining Workshops (ICDMW). IEEE, 2018: 929-936.
> >
> > [3] Zhu M, Weng Y, He S, et al. Towards Graph-hop Retrieval and Reasoning in Complex Question Answering over Textual Database[J]. arXiv preprint arXiv:2305.14211, 2023.

---

### Official Review · Reviewer_gxgg · 2024-11-02

**Soundness:** 3
**Presentation:** 3
**Contribution:** 2
**Rating:** 6
**Confidence:** 4

**Summary:**

This paper introduces FiDeLiS, a retrieval-augmented reasoning framework that enhances LLM performance in KGQA tasks. The framework addresses the challenge of ensuring reliable reasoning by anchoring LLM responses to verifiable reasoning paths within knowledge graphs. FiDeLiS consists of two main components: Path-RAG, which retrieves relevant entities and relations from knowledge graphs using a keyword-enhanced mechanism, and DVBS, which constructs reasoning paths using a combination of natural language planning and beam search. A key innovation is the transformation of path scoring into a deductive reasoning task, moving away from traditional logit-based scoring. Through comprehensive experiments across three datasets, the authors demonstrate that FiDeLiS achieves competitive performance compared to existing baselines while being training-free and computationally efficient. The work contributes to making LLM reasoning more reliable and interpretable in knowledge-intensive tasks.

**Strengths:**

1. The proposed method effectively addresses the reliability issue in reasoning by ensuring each step in the reasoning path can be traced back to the original KG, providing verifiable and interpretable results.
2. The introduction of deductive reasoning verification mechanism offers an innovative solution to the reasoning termination problem, which has been a significant challenge in existing approaches.

**Weaknesses:**

1. There are minor writing issues (e.g., redundant "based" in line 203, "questins" misspelling in line 790) that should be addressed.
2. The paper lacks in-depth analysis of why deductive reasoning verification is more suitable for this task compared to traditional logit-based scoring methods. A theoretical or empirical comparison would strengthen this claim.
3. The core assumption in constructing reasoning paths (that earlier timesteps t have reasoning step candidates St with higher semantic similarity to the query, as reflected in Equation 3) needs more thorough analysis. The paper should investigate whether this assumption holds for problems of varying complexity and whether over-reliance on semantic similarity between reasoning steps and queries might lead to errors.

**Questions:**

- Regarding Algorithm 2, does Path-RAG maintain the same retrieval strategy when obtaining the next possible reasoning step candidates St? Does it incorporate previously formed reasoning steps to aid in retrieval?
- How sensitive is the method to the quality of the knowledge graph? It would be valuable to see an analysis of performance across KGs of varying quality or completeness.
- How does the method handle questions that require commonsense reasoning during the inference process? The current description focuses on constructing reasoning steps from existing KG relations, but real-world questions often require combining structured knowledge with commonsense reasoning.

---

> ### Author Response · Authors · 2024-11-23
>
> We sincerely thank the reviewer for their thoughtful comments. Regarding the writing issues mentioned in **W1**, we have revised the manuscript and corrected the errors. For the other concerns, we provide detailed responses below:
>
> ---
> ### **W2: Why deductive reasoning verification is more suitable compared to traditional logit-based scoring methods?**
>
> Logit-based scoring methods rely on softmax-based probability scores to assess plausibility. These scores lack interpretability and often exhibit overconfidence, where invalid reasoning steps can sometimes be assigned higher probabilities due to inherent model biases. In contrast, our proposed deductive verification follows the idea that each intermediate reasoning step should be **verified** whether it is logically consistent with the previous steps and whether it can contribute to answering the user query (as shown in the example in Lines 987–1020). This mechanism helps minimize the error propagation when extending the reasoning paths, especially for cases when reasoning paths seem plausible but contain subtle errors in intermediate steps.
>
> To validate this claim, we add additional experiment setting using "logit-based scoring", where the endpoint of reasoning path extension was framed as a binary classification task (“yes” or “no”) and we use the corresponding probability scores as the decision criteria. The comparison between deductive reasoning verification and logit-based scoring, as well as adequacy verification (used in ToG), is shown below:
>
> | Methods | WebQSP (hits@1) | CWQ (hits@1) |
> |---|:---:|:---:|
> | FiDeLis + deductive verification | 79.32 | 63.12 |
> | FiDeLis + adequacy verification (used in ToG) | 74.13 | 57.23 |
> | FiDeLis + logit-based scoring | 73.47 | 54.78 |
>
> The results show that deductive reasoning consistently outperforms logit-based scoring by ensuring better logical grounding and reducing overconfidence errors.
>
> ---
> ### **W3: Does Path-RAG perform well across varying complexities, and does it over-rely on semantic similarity?**
>
> We appreciate the reviewer's concern. As shown in Figure 3(a) and (b), Path-RAG consistently achieves a higher coverage ratio of ground-truth reasoning steps compared to baseline across varying complexities of questions (controlled by the reasoning depths required to answer each question). Unlike vanilla retrievers that rely solely on semantic similarity, Path-RAG incorporates structural information via next-hop connections (as shown in Eq. (3)), which significantly enhance the retrieval performance.
>
> To further validate this, we add another baseline from KAPING[1], and report the coverage ratio comparison on CWQ as follows:
>
> | Method | Depth=1 | Depth=2 | Depth>3 |
> |---|:---:|:---:|:---:|
> | Vanilla Retriever | 59.34 | 52.17 | 47.31 |
> | KAPING (top-k triplet retrieval) | 65.72 | 60.41 | 53.11 |
> | Path-RAG | 72.61 | 69.38 | 62.78 |
>
> These results demonstrate that Path-RAG consistently achieves higher coverage, especially for deeper reasoning paths. Unlike the triplet-based retrieval in KAPING, Path-RAG leverages graph structure to capture not only highly relevant nodes and edges but also intermediate “bridge” connecting other highly relevant nodes and edges in next hop neighbors. This design mitigates over-reliance on semantic similarity alone by ensuring that structural relationships are also considered, reducing the likelihood of errors in reasoning path construction.
>
> * [1] Knowledge-Augmented Language Model Prompting for Zero-Shot Knowledge Graph Question Answering: https://arxiv.org/pdf/2306.04136

---

> ### Author Response · Authors · 2024-11-23
>
> ### **Q1: Does Path-RAG maintain the same retrieval strategy when obtaining the next possible reasoning step candidates $S_t$?**
> Yes, the retrieval strategy remains consistent when obtaining the next possible reasoning step candidates $S_t$.
>
> A follow-up question: *"Does it incorporate previously formed reasoning steps to aid in retrieval?"*
>
> By design, we do not explicitly use previously formed reasoning steps during the retrieval process. Instead, this information is utilized in the subsequent LLM reasoning step for enhanced decision-making. The rationale for this design is to keep the retrieval process **lightweight** and **focused** solely on identifying the most relevant next-hop candidates, rather than introducing additional complexity by conditioning on earlier steps. By separating the retrieval and reasoning processes, we aim to balance efficiency and accuracy while allowing the LLM to dynamically integrate information from prior reasoning steps.
>
> ---
> ### **Q2: How sensitive is the method to the quality of the knowledge graph?**
>
> To address the question, we add a new experimental setting where we deliberately manipulated the KG’s quality. Specifically, we perturbed the relations within the KGs to simulate real-world scenarios where some edges may be mislabeled, missing, or incorrectly connected to unrelated nodes. We consider four perturbation heuristics—relation swapping, replacement, rewiring, and deletion to represent the main manifestations of inaccuracies of KG as follows (Find the results in **Appendix H**):
>
> * **Relation swapping** simulates misclassified or mislabeled relationships.
> * **Replacement** introduces spurious links to emulate noise.
> * **Rewiring** reflects structural distortions in graph connectivity.
> * **Deletion** models missing edges or incomplete knowledge.
>
> Our findings, presented in the revised paper (refer to the Appendix H), indicate that the performance of our method remains robust to a reasonable level of perturbation. This robustness is primarily due to our method’s reliance on both semantic similarity and structural information during retrieval, which helps mitigate the effects of incorrect or incomplete edges. Additionally, the LLM’s reasoning capabilities provide further resilience by dynamically compensating for some inaccuracies in the retrieved reasoning paths.
>
> ---
> ### **Q3: How does the method handle questions that require commonsense reasoning during the inference process?**
>
> In this paper, we did not explicitly design mechanisms to handle common sense question answering, as our primary focus is on the KGQA task, where the model answers questions based on knowledge graph (KG) information. However, our proposed method can be adapted to handle commonsense reasoning in the following ways:
>
> (1) **Utilizing Commonsense Knowledge Graphs**: There are existing commonsense knowledge graphs (e.g., ConceptNet) that could serve as valuable resources for providing commonsense knowledge. By integrating such graphs into our framework, the proposed method can construct plausible commonsense reasoning paths to support the question-answering process.
>
> (2) **Leveraging LLMs for Final Reasoning**: Our method relies on LLMs for the final reasoning step, and these models are inherently equipped with fundamental commonsense reasoning capabilities and pre-trained on vast amounts of general knowledge. This enables the LLMs to handle aspects of commonsense reasoning that are not explicitly covered by the knowledge graph.

---

### Official Review · Reviewer_d3aA · 2024-11-04

**Soundness:** 3
**Presentation:** 2
**Contribution:** 2
**Rating:** 5
**Confidence:** 4

**Summary:**

The paper proposes a retrieval augmented reasoning method called FiDeLiS for knowledge graph question answering. The method uses Path-RAG to retrieve relevant entities and relations from KG, and conducts a deductive-reasoning-based beam search to generate multiple reasoning paths leading to final answers. The experiments are conducted on three benchmark KGQA datasets, including WebQuestionSP (WebQSP) , Complex WebQuestions (CWQ) and CR-LT-KGQA, and proves its effectiveness. In addition, the paper also conducts extensive experiments for deep analysis and discussion.

**Strengths:**

Two components of the proposed method, Path-RAG and Deductive-verification Beam Search, are proven to be effective for KGQA.

Extensive experiments are conducted on three benchmarks, including ablation study, analysis experiments and case study.

**Weaknesses:**

The novelty of the paper is still limited. Although the paper proposes two useful components including Path-RAG and DVBS for KGQA, and also demonstrates their effectiveness, however, the main method still follows the paradigm of ToG (Think on Graph).

In the experiments, (1) the important hyper-parameters such as beam width and depth are different when comparing the proposed method and ToG, which will make the comparison unfair. The paper sets the default beam width as 4 and depth as 4, but ToG sets them as 3 in their paper. However, the paper doesn’t mention it. According to Figure 2, ToG would obtain higher performance when setting beam width and depth as 4, although it may be still worse than the proposed method. (2) According to the ablation study in Table 2, replacing Path-RAG with ToG would result in substantial performance declines, the performance would be comparable to or even worse than that of ToG. Does that means the improvement of the method mainly relies on Path-RAG?

Some parts of the paper should be made more clear. For example, after Retrieval in Path-RAG, we can obtain entities $E_m$ and relations $R_m$, and then iteratively construct reasoning step candidates to extend the reasoning paths based them. In addition, DVBS is designed to prompt LLMs to iteratively execute beam search on the reasoning step candidates. Thus, are the reasoning step candidates are all from $E_m$ and $R_m$? does beam search only execute on the candidates?

Some typos and mistakes.

Line 203, two based.

In Figure 2 (d), wrong figure.

In Section 3.4, there is no results for ToG in Table 6, so how to obtain the conclusion that the proposed method shows superior efficiency compared to the ToG.

**Questions:**

See the weaknesses.

---

> ### Author Response · Authors · 2024-11-23
>
> We sincerely thank the reviewer for the thoughtful comments. Below, we address each concern in detail:
>
> ---
> ### **W1: The reviewer is concerned that the novelty of the paper is limited, especially compared with previous work ToG.**
>
> We acknowledge that there are indeed similarities between the proposed method and previous work ToG, especially we both consider using the beam search paradigm for retrieving reasoning paths from KG to guide the LLM reasoning. However, we would like to highlight that our contributions should be not considered as incremental efforts for the following reasons:
>
> **(a) Issues of premature stopping or excessive continuation**: as noted in the manuscript (lines 77 to 81), ToG relies on LLMs to determine when to stop extending a reasoning path by assessing whether the current path is adequate to answer the question. However, this evaluation primarily focuses on superficial relevance and does not ensure that each reasoning step is factually accurate or logically consistent with previous steps. This limitation often results in challenges such as premature stopping or excessive continuation of reasoning paths, leading to retrieved paths that are either incomplete or contain incorrect steps. This issue will hinder the performance of LLMs’ decision making (as shown in the case study from lines 422 to 455).
>
> In contrast, our proposed method uses deductive reasoning as a clear and objective way to decide when to stop extending reasoning paths. Deductive reasoning involves verifying whether each reasoning step logically follows from the previous steps and the user query, making it more reliable and less ambiguous. This approach not only simplifies decision-making but also reduces bias in the reasoning process, ensuring fairer and more accurate stopping criteria. We would like to highlight that this straightforward shift in controlling the end point of reasoning paths demonstrates promising performance, particularly in cases requiring deeper reasoning steps (e.g., CoT and CL-RT with reasoning depths >3), where ToG sometimes struggles to perform effectively.
>
> **(b) Issues of inefficiency in handling large reasoning steps candidates**: In addition, ToG did not consider the retrieval process, which means it considers all neighboring entities/relations during reasoning path extension. This can be particularly problematic for large KGs that have a vast number of neighbors available. It will substantially increase the computational cost for LLM reasoning (as more tokens are considered) and introduces noise from irrelevant nodes and edges, which hampers the effectiveness of subsequent LLM’s decision making process.
>
> In contrast, our method first retrieves entities and relations relevant to the query to construct reasoning path candidates. This allows the LLM to focus on a **smaller, more relevant set of candidates** during decision-making, improving efficiency and reducing the need to filter out irrelevant noise. As demonstrated in our experiments, this retrieval-based approach (path-RAG) consistently achieves better overall performance on different settings (Table 1,2).
>
> Overall, we believe these contributions substantiate the novelty of our work and demonstrate meaningful advancements over ToG, particularly in addressing the key limitations mentioned above.
>
> ---
> ### **W2: Unfair comparison regarding hyper-parameters of beam width and depth.**
>
> Thank you for pointing out this issue. We would like to clarify that the results presented in Table 2 are based on our reproduction of ToG. In these experiments, we ensured that both methods used the same beam width and depth (both set to 4) to maintain a fair comparison. **We have explicitly added this clarification in the revised version of our paper**. Your observation that ToG achieves higher performance with a beam width and depth of 4 is correct, as both methods leverage beam search, and generally, increasing the search space tends to increase the possibility of finding the promising solutions, which could further enhance the overall performance.
>
> ---
> ### **W3: Does the overall performance mainly relies on Path-RAG as shown ablation study in Table 2?**
>
> We appreciate the reviewer's observations regarding the role of Path-RAG in our ablation study in Table 2. While Path-RAG plays a significant role in enhancing the overall performance, we would like to emphasize that the improvements are the result of **synergy between Path-RAG and our deductive reasoning verification mechanism**. As shown in Table 2, replacing Path-RAG with ToG results in a performance decline, but still achieves **1.88% improvement on CWQ** and **0.99% on CR-LT** compared to ToG. It demonstrates that the proposed deductive reasoning perform better on cases requires more complex and multi-step reasoning (where CWQ and CR-LT requires longer reasoning steps) and also verify that the performance enhancement is not solely rely on Path-RAG module.

---

> > ### Comment · Reviewer_d3aA · 2024-11-25
> >
> > Thank the authors for their detailed responses. However, I still have the concerns about the performance of the method.
> > According to Figure 2 in the paper, if the  beam width and depth  are set to 4 for ToG, the Hits@1 is about 60% on CWQ, but the result in Table 1 is only 57.59%.  According to Table 2, when replacing Path-RAG with ToG, the Hits@1 on CWQ is only 59.47%.
> > Similar to WebQSP, if the beam width and depth are set to 4 for ToG, the Hits@1 should be above 75% according to Figure 2, but the value in Table 1 is only 75.13%.
> > By comparing the Table 1 and 2, I still have the concerns on the performance of the method.

---

> > > ### Author Response · Authors · 2024-11-25
> > >
> > > We appreciate the reviewer’s detailed observations regarding the consistency of our results, as maintaining reliability and rigor in our work is a top priority. We would like to highlight that all the experiments reported in the paper were conducted with three independent runs, with the results averaged to mitigate random variations.
> > >
> > > Upon reviewing Figure 2 in light of the reviewer’s comment, we re-checked the experimental logs and identified an anomaly in the CWQ data point of ToG with beam width=4, and depth=4. Specifically, one of the three trials produced an unusually high score, leading to an inflated average in figure 2 (a) and (c) ToG with beam-width=4, and beam-depth=4 (around 60% Hits@1). To validate, we re-ran the experiment using the same configuration and obtained a Hits@1=58.12% at this data point, which generally aligns with the results in Table 1 and 2 which falls within the expected variance range due to stochastic nature of LLMs. **We have corrected this value in Figure 2 (a) and (c), and added a note explaining the adjustment to ensure transparency and prevent further confusion.** We apologize for any oversight and assure it was not intentional. We have also carefully reviewed all other reported results to confirm that this correction does not impact any other findings or conclusions in the paper.
> > >
> > > Regarding the reviewer’s concerns about other ToG’s performance is different in Table 1, Table 2, and Figure 2, we would like to clarify that **these experiments in table1,2 and figure 2 were conducted independently** (to ensure reproducibility under varying configurations). Consequently, minor variations can occur due to factors such as random sampling and LLM stochasticity. However, we would like to highlight these variations are in general within acceptable ranges and do not undermine the overall trends or key conclusions of our work.

---

> ### Author Response · Authors · 2024-11-23
>
> ---
> ### **W4 - Clarification of some minor concerns**
>
> **(1) Are the reasoning step candidates all from $E_m$ and $R_m$? Does beam search only execute on the candidates?**
>
> Yes, you are right. All the reasoning step candidates are constructed from $E_m$ and $R_m$ based on the Eq (3). And the beam search process is executed on the reasoning step candidates at each timestamp.
>
> **(2) How to obtain the conclusion that the proposed method shows superior efficiency compared to the ToG from Table 6.**
>
> We would like to clarify that Table 6 includes results when replacing Path-RAG with ToG for the retrieval mechanism (labeled as “w/o Path-RAG using ToG”). These results show significantly higher average runtime (e.g., 74.26s on WebQSP and 132.59s on CWQ) and increased token usage using ToG compared to our proposed Path-RAG. This directly highlights the efficiency improvements achieved by using Path-RAG.
>
> In addition, on both WebQSP and CWQ, our method consistently achieves lower runtime and token usage while maintaining or improving performance (Hits@1). This demonstrates that our approach streamlines the reasoning process, leading to more efficient use of resources compared to the ToG.

---

### Author Response · Authors · 2024-11-25

We sincerely thank all the reviewers for their helpful comments and suggestions, which have been instrumental in improving our paper. Below is a summary of the major concerns raised and how we addressed them:

---
**(1) Novelty of contributions**: some reviewers raised the concerns that our work is incremental (i.e., ToG) and the novelty is limited. We would like to highlight that while both works build upon the conceptual foundation of reasoning over knowledge graphs (KGs), our work make several original contributions:

- **Enhanced retrieval mechanism (path-rag)**: ToG uses iterative beam search to explore paths in a KG, but its reliance on basic pruning strategies limits the recall and diversity of reasoning paths. FiDeLiS introduces a novel retrieval module, Path-RAG, which leverages an LLM to generate an exhaustive set of query-relevant keywords based on the input query. These keywords are then used to retrieve candidate paths from the KG. This approach increases the likelihood of including potentially relevant paths in the reasoning process (ensuring higher coverage of potential paths) and significantly reduces the chances of missing critical reasoning paths, as demonstrated by our comparative experiments (Tables 1 and 2).

- **Deductive-Verification Beam Search**: Unlike ToG, which relies on standard LLM predictions for pruning, FiDeLiS incorporates **deductive reasoning verification** to validate each reasoning step. This ensures that reasoning paths are logically sound and grounded in the KG, addressing ToG’s vulnerability to misleading paths caused by noisy LLM predictions.

- **Efficiency Without Training**: Like ToG, FiDeLiS is a training-free framework. However, it is computationally more efficient due to the streamlined Path-RAG and DVBS components, which minimize redundant searches while maintaining performance. This makes FiDeLiS more practical for resource-constrained scenarios, as demonstrated by efficiency analysis in Table 6.

**Positioning our work**:
While ToG introduced a significant framework for reasoning over KGs, its scope is constrained by reliance on static beam search and limited error-checking mechanisms. In contrast, our framework starts from **identifying** two fundamental questions when reasoning over KGs: (1) how to retrieve specific knowledge from KG to allow precise reasoning?; (2) how to make the reasoning model understand and utilize the retrieved structured knowledge. To this end, we propose two key modules, Path-RAG and DVBS, which emphasize logical correctness and faithfulness while controlling the efficiency. These modules directly tackle a core limitation of ToG, where reasoning paths often lead to plausible yet unverified steps.

We would like to also mention that incremental advancements are often necessary to solve practical challenges and refine methods. While our work has some overlap with ToG, it addresses crucial gaps in **retrieval recall**, **logical validation**, and **efficiency** issues. These improvements represent a substantial step forward in making KG-enhanced reasoning both more accurate and scalable.

---
**(2) Comparison with Baselines**: some reviewers questioned the fairness of comparisons with ToG and noted the absence of recent baselines and more backbone models:

- **Fair Hyperparameter Settings**: We ensured that both ToG and FiDeLiS used identical beam width and depth settings (set to 4) in all experiments to maintain fairness. This clarification has been explicitly added to the revised manuscript.

- **Integration of Recent Baselines**: While KG-CoT and ToG-2.0 were published contemporaneously with our work, we acknowledge their importance and plan to include comparisons in future work. For GNN-RAG, we clarified that its training-based framework is fundamentally different from our training-free approach and less directly comparable.

- **Additional Backbone Models**: To demonstrate robustness, we extend evaluations to include newer LLMs, such as GPT-4o, Qwen-2-7B, and LLaMA-3.1-8B. The results (e.g., FiDeLiS + GPT-4o achieving 86.34% Hits@1 on WebQSP) underscore the generalizability of FiDeLiS across different architectures.

---
**(3) Role and effectiveness of Path-RAG**: we add new ablations to isolate the Path-RAG’s impact and compare it with semantic-only methods. We observed that Path-RAG outperformed KAPING across varying complexities of questions. While impactful, Path-RAG alone does not account for all performance gains, as deductive reasoning provides complementary benefits as well.

---
**(4) Robustness to KG quality and commonsense reasoning**: we add new experiments to simulate real-world KG inaccuracies through perturbations (e.g., relation swapping, deletion). We observe that FiDelis remains robust and achieves only minor performance drops under a reasonable perturbation level.

---

### Meta-Review · Area_Chair_REyR · 2024-12-23

**Metareview:**

The paper introduces FiDeLiS, a retrieval-augmented reasoning method for knowledge graph question answering (KGQA). It employs Path-RAG for retrieving relevant entities and Deductive-Verification Beam Search (DVBS) for constructing and verifying reasoning paths. While the approach addresses reasoning reliability and efficiency in KGQA, it has substantial weaknesses. The paper lacks theoretical novelty, relying on incremental improvements over prior work such as ToG. The experimental comparisons are incomplete, omitting critical baselines like KG-CoT and ToG 2.0. Additionally, the reliance on ad hoc modules like Path-RAG raises questions about general applicability and scalability. The writing is unclear in several sections, with important methodological details missing or insufficiently explained.

Strengths include addressing a relevant problem, demonstrating empirical improvements on KGQA benchmarks, and releasing code for reproducibility. However, the lack of novelty, incomplete evaluation, and unclear methodology outweigh the contributions.

**Additional Comments On Reviewer Discussion:**

Reviewers raised concerns about limited novelty, unclear methodological choices, and incomplete baseline comparisons. While the authors provided clarifications and additional experiments, key issues remain unresolved. Incremental contributions and missing critical baselines were particularly problematic. The authors’ efforts to address these concerns were appreciated but insufficient to meet the acceptance threshold. The paper’s limitations justify the rejection decision.

---

### Decision · Program_Chairs · 2025-01-22

Reject